# Lifelong regeneration of cerebellar Purkinje cells after induced cell ablation in zebrafish

Sol Pose-Méndez[1]*, Paul Schramm[1], Barbara Winter[1], Jochen C Meier[2], Konstantinos Ampatzis[3], Reinhard W Köster[1]*

[1]Cellular and Molecular Neurobiology, Zoological Institute, Technische Universität Braunschweig, Braunschweig, Germany; [2]Cell Physiology, Zoological Institute, Technische Universität Braunschweig, Braunschweig, Germany; [3]Neuroscience Department, Karolinska Institute, Stockholm, Sweden

**Abstract** Zebrafish have an impressive capacity to regenerate neurons in the central nervous system. However, regeneration of the principal neuron of the evolutionary conserved cerebellum, the Purkinje cell (PC), is believed to be limited to developmental stages based on invasive lesions. In contrast, non-invasive cell type-specific ablation by induced apoptosis closely represents a process of neurodegeneration. We demonstrate that the ablated larval PC population entirely recovers in number, quickly reestablishes electrophysiological properties, and properly integrates into circuits to regulate cerebellum-controlled behavior. PC progenitors are present in larvae and adults, and PC ablation in adult cerebelli results in an impressive PC regeneration of different PC subtypes able to restore behavioral impairments. Interestingly, caudal PCs are more resistant to ablation and regenerate more efficiently, suggesting a rostro-caudal pattern of de- and regeneration properties. These findings demonstrate that the zebrafish cerebellum is able to regenerate functional PCs during all stages of the animal's life.

*For correspondence:
s.pose-mendez@tu-braunschweig.de (SP-M);
r.koester@tu-bs.de (RWK)

**Competing interest:** The authors declare that no competing interests exist.

## Editor's evaluation

Using different zebrafish lines in combination with morphological and functional analysis, the authors provide compelling evidence that ventricular progenitors retains the life-long ability to regenerate PCs. At larval stages the newly regenerated PCs form fully functional circuits that lead to normal behavior. In adult, PC regeneration is less efficient but sufficient to support exploratory behavior. This fundamental study resolves the controversial issue of whether adult PC regeneration is possible and demonstrates that newly formed PCs at larval and adult stages can form functional circuits that supports normal behavior.

## Introduction

Regeneration of neurons in the central nervous system of mammals has been demonstrated to occur, but it is rare and limited to a few brain areas (*Iismaa et al., 2018*). In contrast, in some amphibians and fish, regeneration in almost any part of the central nervous system is evident and occurs not only during development or in juveniles, but also during adulthood (*Kizil et al., 2012*; *Lust and Tanaka, 2019*; *Tanaka and Reddien, 2011*; *Zambusi and Ninkovic, 2020*). Hence, evolutionary conserved fish brain compartments represent a valuable testing ground for characterizing neuronal regeneration in vertebrates.

The zebrafish and mammalian cerebellum share nearly all neuronal cell types with their circuitry, physiology, and function being conserved (*Hashimoto and Hibi, 2012*; *Hibi et al., 2017*; *Koyama et al., 2021*; *Matsuda et al., 2017*; *Volkmann et al., 2010*). Furthermore, clarification of the developmental origin and differentiation program of zebrafish cerebellar neurons, their physiology and functional contribution to locomotor control, motor learning, and socio-emotional behavior has been revealed. For example, like in rodents, Purkinje cells (PCs), together with inhibitory interneurons and eurydendroid cells (ECs) – mammalian deep nuclei neuron equivalents – in zebrafish have been shown to arise from progenitor cells derived from the cerebellar ventricular zone (VZ) expressing the *pancreas-associated transcription factor 1a* (*ptf1a*) (*Kani et al., 2010*; *Kaslin et al., 2013*). In zebrafish, a first wave of postmitotic PCs appears from 3 to 6 days post-fertilization (dpf), afterwards fewer PCs are added slowly over time to achieve a slow continuous increase in PC number aligned with larval growth (*Bae et al., 2009*; *Hamling et al., 2015*; *Namikawa et al., 2019b*). Controversial reports exist about whether in the adult zebrafish cerebellum new PCs are still generated (*Kani et al., 2010*; *Kaslin et al., 2013*; *Kaslin et al., 2009*). Together, these studies have laid a solid foundation for regeneration surveys of cerebellar neurons and PCs in particular (*Chang et al., 2021*; *Chang et al., 2020*; *Harmon et al., 2017*; *Kidwell et al., 2018*; *Knogler et al., 2019*; *Knogler et al., 2017*; *Koyama et al., 2021*; *Markov et al., 2021*; *Matsuda et al., 2017*; *Matsui et al., 2014*; *Rieger et al., 2009*; *Volkmann et al., 2008*).

Based on the extensive adult neurogenesis and regenerative capacity, it is widely believed that all neurons within the zebrafish central nervous system are able to regenerate (*Kroehne et al., 2011*). Yet, in elegant studies this long-held classical view has recently been upset for highly conserved key cerebellar neurons, the PCs (*Kaslin et al., 2017*).

PCs constitute principal information processing neurons of the cerebellum and provide direct output to neurons of the vestibular nuclei (*Knogler et al., 2019*; *Namikawa et al., 2019a*). Hence, PC degeneration in humans results in severe neurological symptoms like ataxia, altered locomotor activity, but also anxiety and social disabilities (*Fancellu et al., 2013*; *Schmitz-Hübsch et al., 2010*). These diseases can be genetically modeled in zebrafish PCs (*Elsaey et al., 2021*; *Hsieh et al., 2020*; *Namikawa et al., 2019a*; *Namikawa et al., 2019b*; *Watchon et al., 2017*), but it has remained elusive so far whether such PC degeneration can be counteracted by regeneration to slow down and mitigate disease progression.

In zebrafish, during embryonic stages, the entire cerebellar primordium including PCs regenerates after surgical removal, due to repatterning processes in the hindbrain (*Köster and Fraser, 2006*). Also, at larval and juvenile stages, the cerebellum recovers from local wounding and regenerates PCs (*Kaslin et al., 2017*). But this PC regeneration capacity declines and is lost at 3 months of age, which has been explained by the exhaustion of proper PC progenitors, while progenitors of other cerebellar neuronal cell types continue to undergo neurogenic divisions (*Hentig et al., 2021*; *Kaslin et al., 2017*). These findings comprehensively suggest that the regeneration capacity of zebrafish PCs is limited, regenerating efficiently during larval and juvenile stages as expected for teleostean neurons, but lacking such a capacity during adulthood resembling mammalian PCs (*Kaslin et al., 2017*).

Yet, larval PC regeneration upon local wounding has been based on the appearance of new PCs, but this could represent ongoing growth of the PC population rather than a regenerative response. In addition, PCs considered to have regenerated cannot be distinguished from non-injured neighbors, therefore the proper physiological integration of regenerated larval PCs into existing circuitry could not be revealed, therefore also in larvae functional PC regeneration is still in question (*Hentig et al., 2021*; *Kaslin et al., 2017*).

Conversely, injuring the adult cerebellum may mask the ability to regenerate PCs. Extensive surgical ablations remove important progenitors and the cellular and extracellular environment likely required for PC regeneration. Local traumatic injuries instead may be counteracted by plasticity of neighboring PCs. In addition, acute traumatic cerebellum injuries affect numerous cell types beyond PCs in a defined area, and if invasive, induce wound healing that could influence regeneration (*Hentig et al., 2021*; *Kaslin et al., 2017*).

Instead, degenerative diseases of PCs affect the entire population of a single-cell type by non-invasive cell death mechanisms (*Meera et al., 2016*). Likely, these principally different situations of PC loss pose different demands and constraints on regeneration. Therefore, the ability of larval and adult PCs to regenerate is still enigmatic.

We have recently introduced transgenic PC-ATTAC zebrafish, in which PCs can be selectively ablated by apoptosis through non-invasive administration of Tamoxifen (*Weber et al., 2016*). This model combines several advantages to address the open question of the occurrence of functional PC regeneration: (1) Ablating nearly the entire PC population uncouples plasticity from regeneration and creates a high regenerative demand; (2) PC death by apoptosis and microglia phagocytosis (*Weber et al., 2016*) prevents leakage of PC contents into the extracellular environment and resembles disease-associated neurodegeneration and; (3) Fluorescent protein expression in PCs allows not only for quantifying ablation and recovery efficiencies, but also for investigating physiological integrity of regenerating PCs (*Hsieh et al., 2014*) and relating behavioral recovery to the extent of PC population regrowth (*Champagne et al., 2010*; *Huang and Neuhauss, 2008*); (4) Age-independent Tamoxifen treatment allows to study PC regeneration in larvae, juveniles, and adults under equivalent conditions. Using this PC-specific ablation approach, we have readdressed the open question of PC regeneration by combining in vivo imaging, electrophysiology and behavioral analysis in both the larval and the adult zebrafish cerebellum.

## Results

### Efficient recovery of larval PCs after induced non-invasive ablation

Carriers of the stable transgenic zebrafish strain PC-ATTAC express a membrane-targeted red fluorescent protein (FyntagRFP-T) as a reporter together with a Tamoxifen-inducible Caspase 8 selectively in postmitotic cerebellar Purkinje neurons (*Figure 1A*; *Weber et al., 2016*). Between 4 and 6 days post-fertilization (dpf) the PC layer in zebrafish is established and matures (*Hamling et al., 2015*; *Namikawa et al., 2019b*). Incubation of PC-ATTAC larvae between 4 and 6 dpf with 4-hydroxy-tamoxifen (4-OHT) overnight induced apoptosis in the majority of PCs (85–95%) leaving red fluorescent apoptotic debris behind, which progressively disappeared by microglial phagocytosis within 3–5 days post-treatment (dpt) (*Weber et al., 2016*). Since ethanol (EtOH) was used as a solvent for 4-OHT stock solutions, EtOH-treated (0.4% final concentration) specimens were used as controls, which did not show any sign of PC death (*Figure 1B*).

About 5–7 dpt first red fluorescent PCs reappeared, and already after about 11 dpt nearly half of the PC population compared to controls was replenished (*Figure 1B, C*). This demonstrates that during larval stages the PC population can quickly recover from a nearly complete removal.

### Ablated larval PC population regenerates completely

The partial PC regeneration observed within a few days could either represent a regenerative response, the continuation of PC layer development or a combination of both. Only in the latter case the PC population would regain its full size compared to controls. Therefore, PC numbers were quantified from 1 to 6 months post-treatment (mpt) (*Figure 1D–G*). Compared to PCs found in non-ablated controls, numbers of PCs in specimens after initial PC ablation reached 70% at 1 mpt, 91% at 3 mpt, and 100% at 6 mpt, respectively (*Figure 1D–F*). Since PCs ablated during larval development (ca. 400) amount to about one quarter of the adult PC population at 6 months (ca 1500), an obvious reduction in PC population size would be expected, if the ablated PCs were not replaced, but such a reduction in PC numbers was not observed. Instead, a full recovery of PCs was evident at 6 mpt.

In addition, no apparent anatomical differences of the cerebellar cortex at 1 and 6 mpt could be observed by immunohistochemistry against the PC-specific antigen ZebrinII on sagittal sections from PC-ablated and control specimens (*Figure 1G*). In addition, quantification of the distance between anterior and posterior PCs in the PC layer on sagittal sections through the corpus cerebelli (CCe) of PC-ablated and control specimens at 1, 3, and 6 mpt revealed no significant differences (*Figure 1—figure supplement 1A–C*). These findings corroborate the complete restoration of the ablated larval PC population (*Figure 1D–G*).

To obtain further insight into the temporal dynamics of PC replacement, proliferating cells were labeled by Bromodeoxyuridine (BrdU) treatment for 24 hr at different time points post-ablation followed by their immunohistochemical detection and quantification. During the first 7 days post-ablation no significant increase in proliferating cell numbers could be observed in PC-ablated larvae compared to EtOH-treated controls (*Figure 1—figure supplement 2A–D*). The same result was obtained when this BrdU treatment was repeated around 2 weeks later at 18 dpt (*Figure 1—figure supplement 3A–G*).

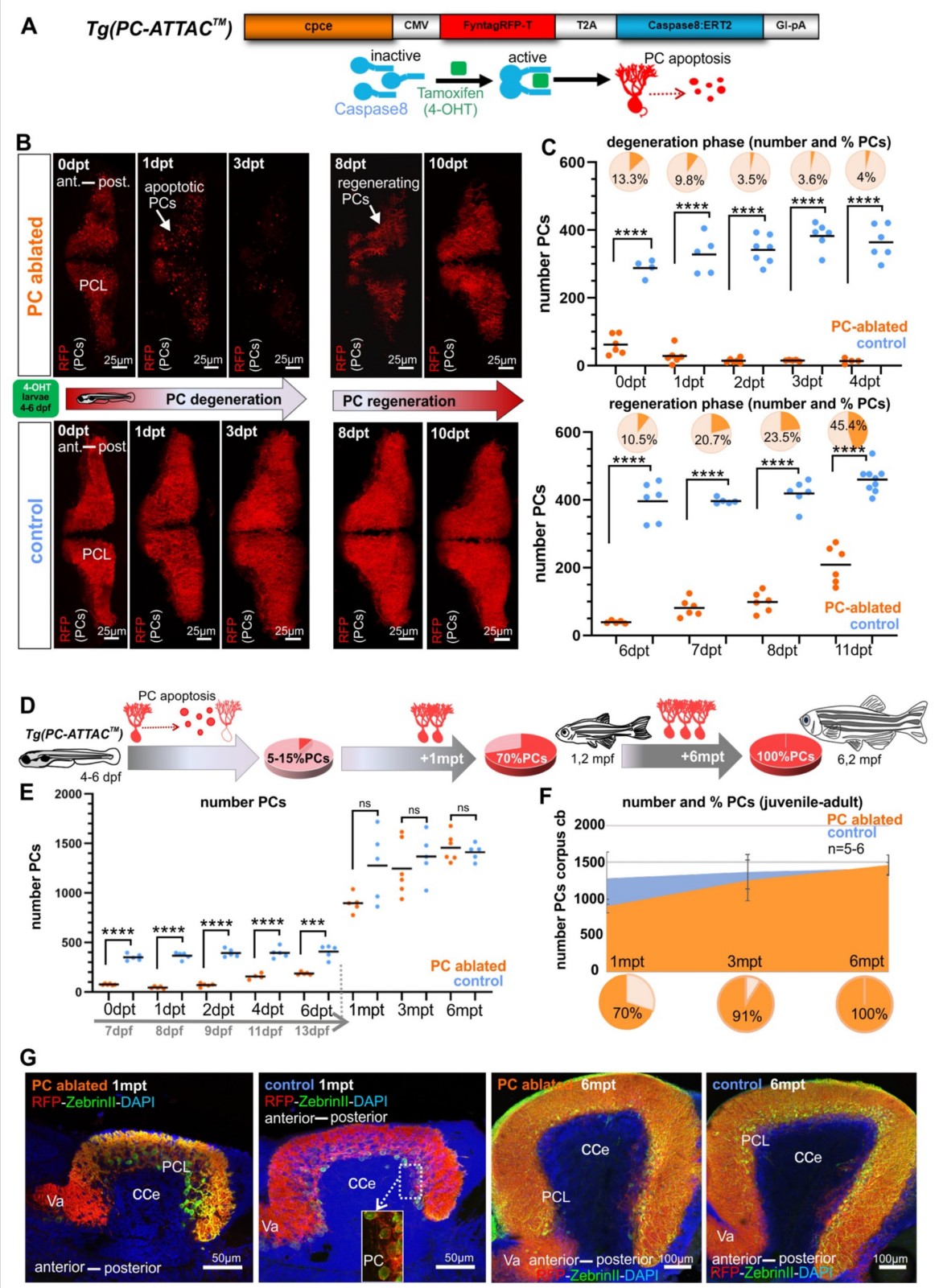

**Figure 1.** Induced Purkinje cell (PC) ablation in zebrafish larvae: time course of degeneration and regeneration. (**A**) Construct used to generate the Tg(PC-ATTAC) transgenic line, modified from *Weber et al., 2016*. (**B**) Images of the PC layer (fyn-tagRFP-T fluorescence expressed in mature PCs) after induced PC ablation in larvae at 4–6 dpf monitoring 10 days after 4-hydroxy-tamoxifen (4-OHT) treatment. (**C**) Number and percentage of PCs in ablated (4-OHT) vs control (EtOH) larvae. (**D–G**) Quantitative monitoring of PC regeneration after induced PC ablation in larvae. (**D–F**) Numbers and

*Figure 1 continued on next page*

*Figure 1 continued*

percentage of PCs until 6 mpt. (**G**) Images of PCs on sagittal cerebellar sections after immunostaining with anti-tagRFP and anti-ZebrinII antibodies, comparing ablated and control groups at 1 and 6 mpt. Statistical information: sample size n=4-9, statistical method=unpaired t-test two tailed, levels of significance=P <0.0001 (****), P=0.0001 (***). Additional information in **Supplementary file 1**.

The online version of this article includes the following source data and figure supplement(s) for figure 1:

**Source data 1.** Purkinje cell (PC) quantification for *Figure 1C, D, F*.

**Figure supplement 1.** Quantification of the distance between rostral and caudal Purkinje cell (PC) layer in the corpus cerebelli of juveniles and adults after PC ablation in larvae, related to *Figure 1*.

**Figure supplement 1—source data 1.** Antero-posterior distance measurements of Purkinje cell (PC) layer for *Figure 1—figure supplement 1A*

**Figure supplement 2.** Cell proliferation in the cerebellum after Purkinje cell (PC) ablation: Bromodeoxyuridine (BrdU) analysis after PC ablation in larvae, related to *Figure 1*.

**Figure supplement 2—source data 1.** Numbers of BrdU$^+$ cells for *Figure 1—figure supplement 2D*.

**Figure supplement 3.** Bromodeoxyuridine (BrdU)/5-Ethynyl-2'-deoxyuridine (EdU) double pulse chase after Purkinje cell (PC) ablation in larvae, related to *Figure 1*.

**Figure supplement 3—source data 1.** Numbers of Purkinje cells (PCs) for *Figure 1—figure supplement 3F* and of double-positive BrdU$^+$ and EdU$^+$ PCs for *Figure 1—figure supplement 3G, H*.

A second pulse of 5-Ethynyl-2'-deoxyuridine (EdU) 4 days later at 22 dpt revealed a much lower number of BrdU/EdU double-positive cells at 26 dpt, indicating that most of the newly generated PC progenitors passed on to differentiation. However, in PC-ablated specimens the number of BrdU/EdU double-positive cells was significantly elevated suggesting that PC progenitors add on average more PCs to the PC layer by additional rounds of proliferation in PC-ablated zebrafish (*Figure 1—figure supplement 3A–H*). Together these findings show that PC regeneration was neither initiated nor later promoted by a burst in cell proliferation, but occurred continuously by slowly adding surplus PCs over time via a delayed cell cycle exit until the full PC population size was reestablished 6 months later.

## Neuroepithelial *ptf1a*-expressing cells are progenitors of regenerating PCs

Since radial glia is considered one of the potential sources for neuronal regeneration, we crossed carriers of the transgenic Tg(*gfap*:EGFP) strain with green fluorescent radial glia cells into the PC-ATTAC background. Yet, after PC ablation none of the reappearing PCs expressed both fluorescent reporters (*Figure 2A*), nor did the radial glia cell population increase significantly (*Figure 2B*). Immunohistochemistry in control larvae using an antibody against brain lipid-binding protein (Blbp) expressed in radial glia showed coexpression in 90% of *gfap*:GFP-positive radial glia, but also failed to detect Blbp expression in PCs (*Figure 2—figure supplement 1*). This makes the contribution of radial glia to larval PC differentiation and regeneration unlikely, although it cannot be excluded that *gfap*-enhancer-driven EGFP expression had been lost by the time regenerating PCs differentiate.

As an alternative, VZ-derived neuroepithelial cells were tested that give rise to GABAergic cerebellar neurons. These VZ-derived progenitors express the transcription factor Ptf1a and among others give rise to PCs in mammals as well as in zebrafish (*Hoshino et al., 2005*; *Kani et al., 2010*). Hence, transient green fluorescent protein expression in the transgenic strain Tg(*ptf1a*:EGFP) can be used for monitoring these VZ-derived progenitor cells (*Bae et al., 2009*). Indeed, double fluorescent PCs in Tg(*ptf1a*:EGFP) larvae in the PC-ATTAC background were observed following PC ablation (*Figure 2C–E*) and were significantly increased in their ratio compared to only red fluorescent PCs for about 2 weeks starting two days after ablation (*Figure 2F, G*).

The ratio of double fluorescent PCs is in favor of newly differentiated PCs, as EGFP expression by the *ptf1a* enhancer is only transient. Mature PCs in control specimens therefore, exceed mature PC numbers in larvae with active PC regeneration (*Figure 2—figure supplement 2A*). The absolute number of *ptf1a*:EGFP-expressing cells in the cerebellum was not significantly higher in PC-ablated PC-ATTAC larvae compared to controls (*Figure 2—figure supplement 2B*) consistent with the BrdU labeling results. In addition, a small increase in green fluorescent PC progenitor cells may be further masked as *ptf1a*:EGFP-expressing cells contribute to several neuronal lineages of the cerebellum (*Bae et al., 2009*). In conclusion, larval PC regeneration is driven by a small increase of VZ-derived *ptf1a*-expressing neuroepithelial cells, the natural PC progenitor population.

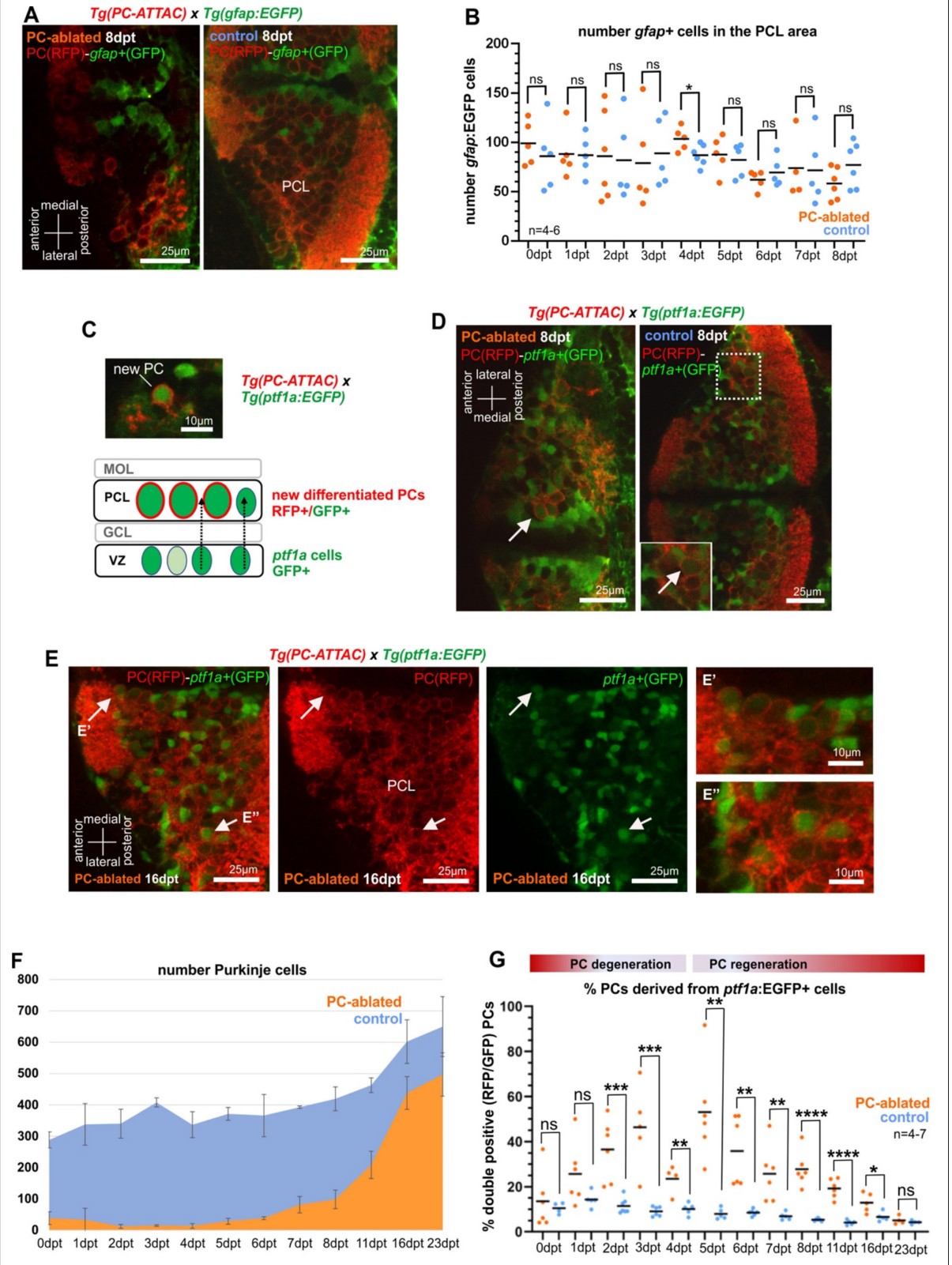

**Figure 2.** Cellular analysis of potential progenitors of regenerating Purkinje cells (PCs). (**A**) Images of larval cerebellum of the double transgenic line Tg(PC-ATTAC)/Tg(*gfap*:EGFP) after induced PC ablation. (**B**) Number of gfap+ cells throughout the PC layer area during PC degeneration and beginning of regeneration. (**C**) Illustration of new PC development from *ptf1a*+ progenitors. (**D, E**) Images of larval cerebellum of the double transgenic line Tg(PC-ATTAC)/Tg(*ptf1a*:EGFP) after induced PC ablation, revealing double-positive cells (arrows). Average of PC numbers (**F**) and percentage of

*Figure 2 continued on next page*

*Figure 2 continued*

PCs showing GFP fluorescence double-positive cells, (**G**) during degeneration and regeneration of PCs. The red fluorescent protein from the PC-ATTAC strain is exclusively expressed in the cell membrane, while EGFP from the *ptf1a*- and *gfap*-reporter strains localizes to the cytoplasm. GFP and RFP were enhanced by fluorescence immunohistochemistry in **A, C–E**. Statistical information: statistical method=unpaired t-test two tailed, levels of significance=P<0.05 (*), P<0.01 (**), P<0.001 (***), P<0.0001 (****). Additional information in *Supplementary file 1*.

The online version of this article includes the following source data and figure supplement(s) for figure 2:

**Source data 1.** Quantification of *gfap*-expressing cells for *Figure 2B* and quantification of *ptf1a*:GFP-expressing PC-ATTAC cells for *Figure 2G*.

**Figure supplement 1.** Identity of glial cells in the larvae cerebellum: images of double staining with radial glial markers and cell precursors and specific cell type markers, related to *Figure 2*.

**Figure supplement 1—source data 1.** Percentage of double-positive ptf1a:GFP/antiBLBP cells.

**Figure supplement 2.** GABAergic cell progenitors in the cerebellum after Purkinje cell (PC) ablation, related to *Figure 2*.

**Figure supplement 2—source data 1.** Quantification of ptf1a:GFP-expressing cells for *Figure 2—figure supplement 2A, B*.

While radial glia could act as progenitors of *ptf1a*:GFP-positive VZ-derived neurons and therefore also of regenerating PCs, only a small fraction of *ptf1a*:GFP cells of about 10% were found to coexpress Blbp, with an even smaller fraction of *olig2*:GFP-positive cells displaying coexpression of Blbp, which likely present *ptf1a*-derived progenitors of oligodendrocytes (*Figure 2—figure supplement 1*). Hence, progenitors of regenerating PCs are unlikely to be derived from radial glia-expressing *gfap* or Blbp or have lost *gfap*-enhancer-driven EGFP expression despite the long half-life of the GFP protein (*Li et al., 1998*) at the time of PC differentiation. Unambiguous clarification of the early progenitors of regenerating PCs has to await precise genetic fate mapping tools for permanently tracing radial glia derivatives in the zebrafish cerebellum.

Interestingly, both double fluorescent recently differentiated PCs (*Figure 2D, E*) and regenerating PCs marked by cumulative BrdU labeling (*Figure 2—figure supplement 2C*) were not randomly distributed, but clustered loosely in two domains along the dorsal midline and in lateral parts of the cerebellum, respectively. This suggests that the VZ is spatially patterned like the adjacent germinal zone of the upper rhombic lip (*Volkmann et al., 2008*).

## Regenerating PCs reestablish the physiology of cerebellar circuitry

To obtain a reference for circuit reestablishment by regenerating PCs, we first analyzed the electrophysiological properties of PCs in 4- to 21-dpf-old larvae of the PC reporter line Tg(–7.5ca8:GFP)[bz12] under resting conditions by patch-clamp recordings in attached cell configuration (*Figure 3A–E*). These studies confirmed that differentiating PCs require about two days to establish a stable firing pattern of a mature PC with an average firing frequency between 8 and 10 Hz, displaying high-frequency bursts of over 30 Hz and a ratio for complex spikes to simple spikes (cs/ss) of 0.04 (*Harmon et al., 2017*; *Hsieh et al., 2014*; *Scalise et al., 2016*; *Sengupta and Thirumalai, 2015*). First PCs born from 4–6 dpf forming the initial PC layer reach a plateau of electrophysiological maturity around 8 dpf (*Figure 3—figure supplement 1A–C*).

Next, reappearing PCs after ablation in PC-ATTAC larvae in the Tg(–7.5*ca8*:GFP)[bz12] background with ablation rates above 90% were investigated (*Figure 3A-E*, *Figure 3—figure supplement 1D-G*). During the acute phase of PC degeneration and debris clearance (0–4 dpt), the average frequency and the bursting frequency of few remaining PCs in PC-ablated larvae were significantly lower compared to controls and also the cs/ss ratio was elevated in an obvious trend (*Figure 3—figure supplement 1E–G*). During the acute regeneration phase (5–10 dpt) increasing numbers of PCs displayed characteristic simple and complex spike patterns (*Figure 3B, B'*) with the average tonic firing and spontaneous burst frequencies steadily increasing to normal mature PC values. Similarly, the cs/ss ratio after an initial increase, reestablished ratios of mature PCs. During the extended regeneration period (10–21 dpt) the average tonic firing frequency and cs/ss ratio remained stable and compared well to mature control PCs. This indicates that regenerating PCs successfully integrated into the remaining cerebellar network and regenerated PCs were indistinguishable from controls at 10 dpt. Of note, the average spontaneous burst frequency of regenerated PCs was increased by 1.5-fold in regenerated PCs compared to controls at 21 dpt (*Figure 3D*), suggestive of a possible compensatory mechanism (see discussion).

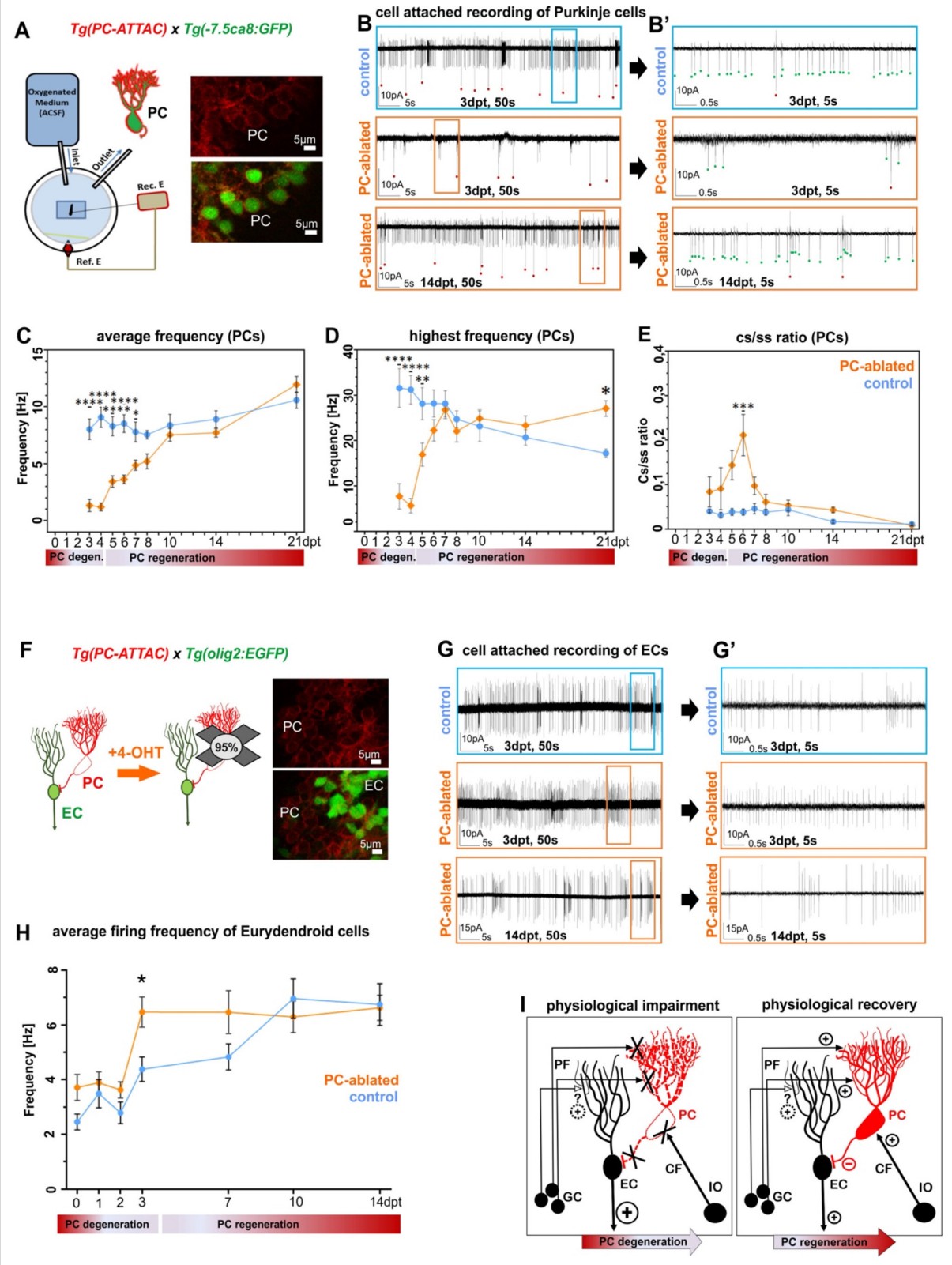

**Figure 3.** Electrophysiological properties of Purkinje cells (PCs) and eurydendroid cells (ECs) during ablation and regeneration phase. (**A**) Patch-clamp recording setup used for all experiments. Fluorescent PCs in larvae of the double transgenic line Tg(PC-ATTAC)/Tg(–7.5*ca8*:EGFP). (**B**) 50 s trace of representative recordings of the tonic firing activity in control larvae 3 days after EtOH or in PC-ablated larvae 3 and 14 days after 4-hydroxy-tamoxifen (4-OHT) treatment and (**B′**) 5 s traces for all three traces shown. Red dots mark complex spikes and green dots mark simple spikes. (**C–E**) Diagrams

*Figure 3 continued on next page*

*Figure 3 continued*

representing results of the electrophysiologic investigations in PCs. (**C**) Average tonic firing frequency plotted vs dpt. (**D**) Highest spontaneous bursting frequency over an interval of 1 s during a 100 s trace plotted vs dpt. (**E**) Ratio of the complex spikes to simple spikes vs dpt. (**F**) Illustration of PC loss after 4-OHT treatment and representative image of the PC layer from a double transgenic Tg(PC-ATTAC)/Tg(*olig2*:EGFP) larva. (**G**) 50 s trace of representative recordings of the tonic firing activity of ECs in control larvae 3 days after EtOH or in PC-ablated larvae 3 and 14 days after 4-OHT treatment and (**F**) 5 s traces for all 3 traces shown. (**H**) Average tonic firing frequency of ECs after PC ablation plotted vs dpt. (**I**) Illustration of physiological impairment and recovery model of the input/output of PCs during PC degeneration and regeneration, respectively. Statistical information: sample size n=17-45, statistical method=2-Way ANOVA, Šídák's multiple comparisons test, levels of significance=P<0.05 (*), P<0.01 (**), P<0.001 (***), P<0.0001 (****). Additional information in *Supplementary file 1*.

The online version of this article includes the following source data and figure supplement(s) for figure 3:

**Source data 1.** Average firing frequency determination of Purkinje cells (PCs) for *Figure 3C*, highest burst frequency numbers of PCs for *Figure 3D*, and dataset of complex spike to simple spike firing ratios of PCs for *Figure 3E*.

**Figure supplement 1.** Reference set of physiological properties of Purkinje cells (PCs) and physiological properties of individual PCs after ablation, related to *Figure 3*.

**Figure supplement 1—source data 1.** Average firing frequency determination of Purkinje cells (PCs) for *Figure 3—figure supplement 1A*, highest burst frequency numbers of PCs for *Figure 3—figure supplement 1B*, and dataset of complex spike to simple spike firing ratios of PCs for *Figure 3—figure supplement 1C*.

**Figure supplement 2.** Purkinje cell (PC) activity recovery after repeated cell type-specific ablation, related to *Figure 3*.

**Figure supplement 2—source data 1.** Quantification of Purkinje cells (PCs) for *Figure 3—figure supplement 2A*, and PC activity for *Figure 3—figure supplement 2B-D*.

**Figure supplement 3.** *Olig2*-positive cells after Purkinje cell (PC) ablation in larvae, related to *Figure 3*.

**Figure supplement 3—source data 1.** Numbers of olig2:GFP-expressing cells at 15 dpt for *Figure 3—figure supplement 3B*.

**Figure supplement 4.** Eurydendroid cell (EC) activity recovery after Purkinje cell (PC) ablation, related to *Figure 3*.

**Figure supplement 4—source data 1.** Quantification of Purkinje cells (PCs) for *Figure 3—figure supplement 4A*, and eurydendroid cell (EC) activity for *Figure 3—figure supplement 4B-D*.

**Figure supplement 5.** Axon tracing of regenerating Purkinje cells (PCs), related to *Figure 3*.

As about 5–10% of the PCs are able to escape Tamoxifen-mediated apoptotic ablation, it could still be possible that this small fraction reestablishes PC physiology by means of plasticity, while the newly added regenerating PCs fail to integrate into the cerebellar circuitry properly. In this case the potential ablation resistant PC fraction and their physiological pattern should remain unaffected by a second ablation as rewiring by plasticity has been established. Yet, such a second Tamoxifen treatment at 10 dpt when about half of the PC population had been reestablished, resulted again in the reduction of the PC population to about 6% (*Figure 3—figure supplement 2A*). Physiological measurements revealed – similar to the first ablation period – a significant decrease in average firing and bursting frequency and a significant increase in the cs/ss ratio of the remaining PCs. Within the following 10 days the PC population recovered from the repeated ablation to about 64% compared to PC numbers in age-matched control larvae, and the physiological signatures of the average firing frequency and cs/ss ratio returned to values indistinguishable from non-ablated PCs in control specimens (*Figure 3—figure supplement 2*). These findings argue strongly against a physiological recovery of the ablated PC population solely by plasticity mechanisms and indicate that this recovery is majorly driven by newly added regenerating PCs that properly reintegrate into the cerebellar circuitry.

ECs are the principal direct efferences of zebrafish PCs and represent the neuronal equivalent of deep cerebellar nuclei neurons in mammals (*Heap et al., 2013*; *Matsui et al., 2014*). PC-ATTAC carriers were crossed into the Tg(*olig2*:EGFP)[vu12] background, in which ECs are marked by EGFP expression (*Figure 3F*; *McFarland et al., 2008*), to analyze the consequences of PC ablation and regeneration for EC physiology. When green fluorescent ECs were quantified no difference in their number between PC-ablated and control larvae could be observed suggesting that ECs in the cerebellum of PC-ablated larvae are generated in appropriate numbers (*Figure 3—figure supplement 3A, B*). However, at 3 dpt the average firing frequencies and highest burst frequencies of ECs were significantly elevated upon acute loss of PC inhibitory input (*Figure 3—figure supplement 4B–D*), as ECs may still receive excitatory input from parallel fibers (*Hashimoto and Hibi, 2012*; *Hibi and Shimizu, 2012*). This elevated firing frequency returned already at 7 dpt to a slight non-significant elevation during ongoing PC regeneration, and was indistinguishable from ECs in controls from 10

dpt onwards (*Figure 3—figure supplement 4B–D*). This suggests that regenerating PCs quickly reestablish proper inhibitory input with their direct efferences (*Figure 3I*). Injection of a bicistronic 4xcpce vector (*Namikawa et al., 2019b*) driving expression of the nuclear located histone2B-tagBFP2 (H2B-tagBFP2) and the membrane-associated mCardinal-CAAX fluorescent reporter proteins into dual transgenic PC-ATTAC/*olig2*:GFP one cell-stage embryos resulted after PC ablation in some regenerating PCs in which mCardinal-fluorescent membrane protrusions from these regenerating PCs with stronger expression in axons – probably mediated by the C-terminal palmitoylation motif (*Steele-Nicholson and Andrews, 2022*) – could be determined by confocal microscopy adjacent to ECs sometimes wrapping around their somata at 10 dpt (*Figure 3—figure supplement 5A–D*). These findings support a fast reestablishment of inhibitory PC to EC neurotransmission by regenerating PCs. Alternatively, ECs adapt in their intrinsic firing properties to the lack of inhibitory PC input to compensate for an excessive parallel fiber excitation. Further mechanistic studies are needed once further neuroanatomical evidence for likely direct synaptic contacts between parallel fiber and ECs have been provided.

## Behavioral analysis demonstrates functional recovery of cerebellar circuitry by larval PC regeneration

To address visuo-motor behavior, PC-ATTAC larvae with a PC-ablation efficiency of above 90% (*Figure 4—figure supplement 1A*) were mounted in agarose to evaluate cerebellum-dependent optokinetic response (OKR) performance (*Namikawa et al., 2019b*; *Huang and Neuhauss, 2008*). Right eye rotations induced by horizontally moving stripes and fast return of the eye (saccades) were recorded as deflection angle with respect to the body axis (*Figure 4A, B*). At 2 dpt, when PC degeneration is prominent, the speed of eye movement during object tracking and saccade frequency were reduced compared to controls (*Figure 4C left panel, D*). Strikingly, at 10 dpt when on average 29% of PCs had been reestablished (*Figure 4—figure supplement 1A*), OKR response had recovered neither differing in eye movement speed nor in saccade frequency compared to controls (*Figure 4C right panel, D*).

In order to evaluate cerebellum-dependent locomotor behavior, the swimming of individual control or PC-ablated PC-ATTAC larvae were tracked in a 12-well plate for 6-min intervals (*Figure 5A*). Subsequently, PC ablation efficiency by counting PCs was determined to range above 90% (*Figure 4—figure supplement 1B*). As EtOH treatment (the solvent for 4-OHT) can cause hyperactivity, while 4-OHT treatment is known to suppress hyperactivity (*Blaser et al., 2010*; *Hoffman et al., 2016*), respective solvent and compound controls were included. Also untreated and 4-OHT-treated wild-type larvae (to exclude effects from PC-ATTAC transgene expression) were analyzed, with the latter considered the most appropriate control. At 2 dpt PC-ablated larvae traveled longer distances at elevated mean and maximum swim speeds suggesting hyperactive behavior, even above EtOH-treated controls (*Figure 5D upper row, E*). Between 9 and 11 dpt, when PC replenishment ranged on average around 40% (*Figure 4—figure supplement 1B*), all swim parameters had recovered and were not different to all control groups, respectively (*Figure 5D lower row*).

To analyze socio-emotional behavior, swim patterns in the above experiments were reevaluated for thigmotaxis. At 2 dpt PC-ablated specimens spent significantly more time swimming along the walls and less frequently entered the open zone of the arena (*Figure 5B, C upper row*). Such an increase in thigmotaxis is considered to relate to elevated anxiety (*Richendrfer et al., 2012*). Again, at 9–11 dpt PC-ablated specimens displayed no significant differences in thigmotaxis compared to all control groups (*Figure 5B, C lower row*).

Together these findings from behavioral analysis show that PCs not only reestablish the proper physiological signature after regeneration, but also regain their functional properties within 10 days in controlling visuo-motor, locomotor and thigmotactic functions already at a PC recovery rate around 50% (*Figure 5F*).

## Significant recovery of adult PCs after induced ablation

The currently held view is that zebrafish PCs are only able to regenerate during larval stages, while this ability ceases in juveniles. Adult sexually mature zebrafish older than 3 months post-fertilization (mpf) are considered unable to replace lost PCs, like the mammalian cerebellum (*Kaslin et al., 2017*). The

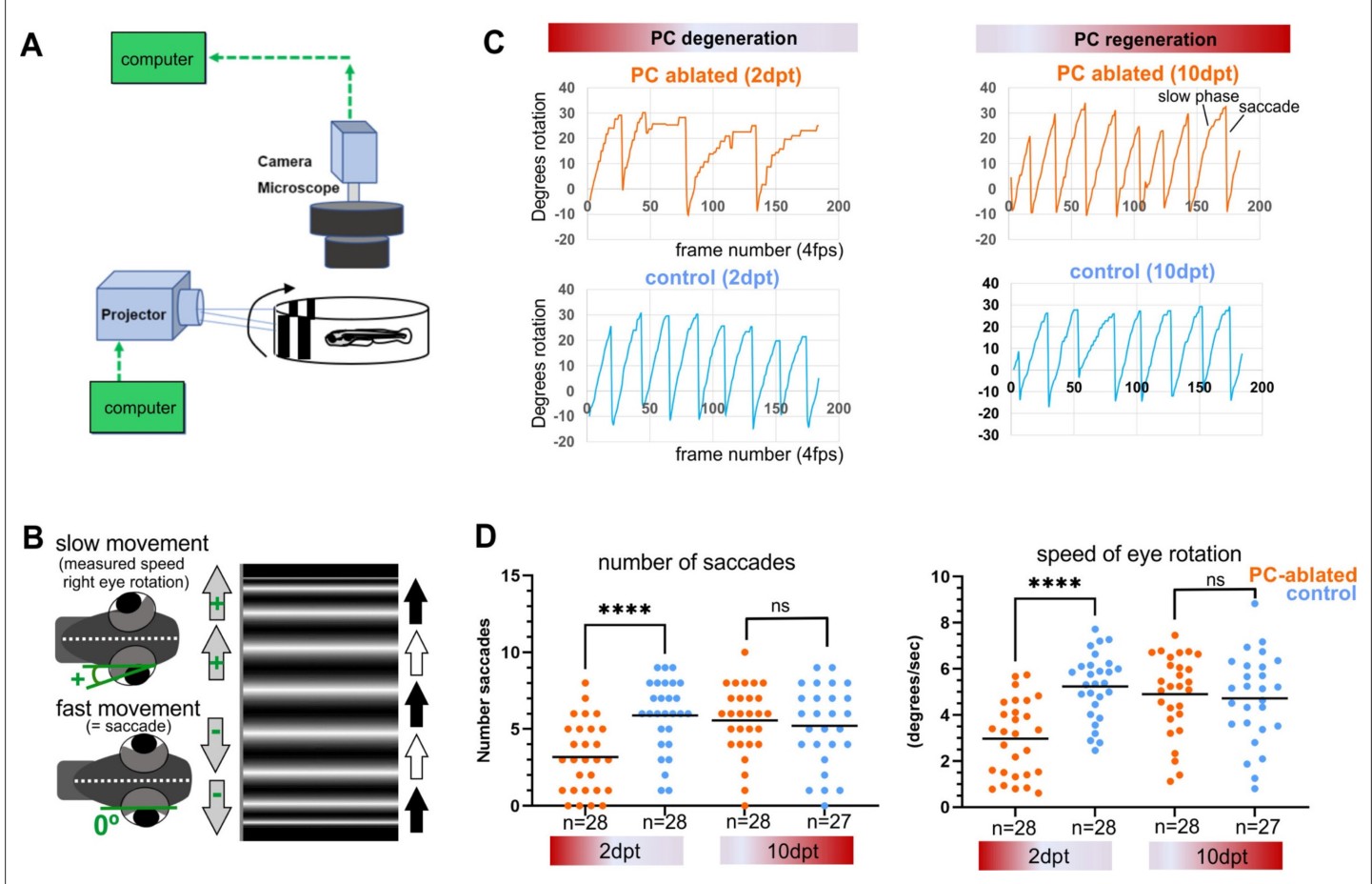

**Figure 4.** Visuo-motor behavior analysis: optokinetic response (OKR) after induced Purkinje cell (PC) ablation in larvae. Illustration of OKR setup (**A**) and eye movements in larvae (**B**) during performance of the OKR test. (**C**) Representative graphs showing OKR response performance during acute PC degeneration (2 dpt) and regeneration (10 dpt) phases in PC-ablated (4-hydroxy-tamoxifen [4-OHT] treated) vs control group (EtOH treated). (**D**) Quantification of OKR response: number of normal saccades (eye rotation >19.5°) and speed of eye rotation during slow phase movements at 2 and 10 dpt. The data correspond to the results of two independent trials that were pooled. Statistical information: statistical method=Mann-Whitney test or unpaired t-test, two tailed, level of significance=P<0.0001 (****). Additional information in ***Supplementary file 1***.

The online version of this article includes the following source data and figure supplement(s) for figure 4:

**Source data 1.** Datasets of eye deflection measurements during optokinetic response (OKR) behavior for ***Figure 4C, D***.

**Source data 2.** Datasets of eye deflection measurements during optokinetic response (OKR) behavior for ***Figure 4C, D***.

**Source data 3.** Eye rotation and saccade quantification for ***Figure 4C, D***.

**Figure supplement 1.** Average percentage of Purkinje cells (PCs) at acute degeneration phase and beginning of regeneration from independent trials of optokinetic response (OKR) (**A**) and free-swimming tests (**B**) after PC ablation in larvae, related to ***Figures 4 and 5***.

**Figure supplement 1—source data 1.** Quantification of Purkinje cells (PCs) for ***Figure 4—figure supplement 1A, B***.

efficient functional regeneration of ablated PCs during larval stages lasting until adulthood prompted us to challenge this current view of limited cerebellar plasticity.

We therefore, induced apoptotic cell death in heterozygous PC-ATTAC adults at 5 months of age (***Figure 6A***) by three consecutive treatments with Endoxifen, an active metabolite of 4-OHT, which resulted in higher PC ablation efficiencies in adults compared to 4-OHT. As apoptosis of PCs and clearance of debris took longer than in larvae, remaining PCs were quantified on consecutive vibratome sections of the CCe at 13–14 dpt (***Figure 6C, F***), which revealed an ablation efficiency above 90% compared to solvent controls (***Figure 7A, B, I***). Interestingly, while in the rostral parts of the CCe PC ablation was nearly complete, PCs in the caudal part were more resistant to apoptotic ablation (***Figure 6A, C, F***).

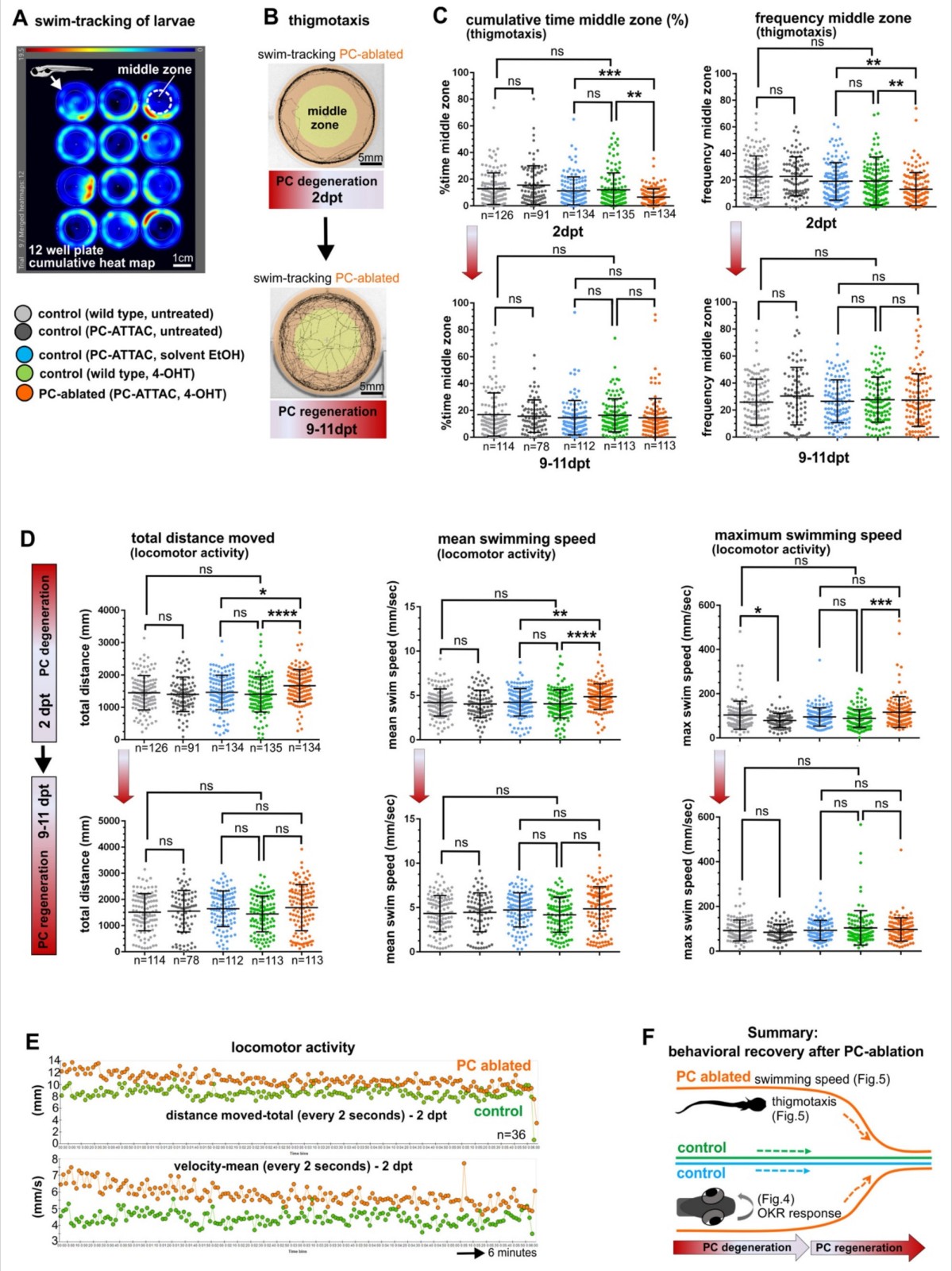

**Figure 5.** Swimming behavior analysis after induced Purkinje cell (PC) ablation in larvae. (**A**) Heat map representing the location of zebrafish larvae in a 12-well plate during 6 min of swimming. (**B**) Example of swim track after PC ablation (2 dpt) and during regeneration (10 dpt) phases. (**C**) Quantitative analysis of swim preferences along the edge vs center zone of the arena (frequency of visits and percentage of time spent in the center zone) in PC-ablated larvae vs control groups. (**D**) Quantitative analysis of locomotor activity (total distance traveled, mean and maximum swimming speed) in PC-

*Figure 5 continued on next page*

Figure 5 continued

ablated larvae vs controls. (**E**) Graphs showing distance traveled and swim speed every 2 s during the tracking period, comparing PC-ablated vs control larvae. (**F**) Illustration summarizing impairment and recovery of optokinetic response (OKR) (**Figure 4**) and locomotor behavior during PC degeneration and early regeneration phases, respectively.The data from free swim tests correspond to the results from four independent ablations that were pooled. Statistical information: size sample n=78-135, statistical method=ANOVA Kruskal-Wallis, Dunn´s multiple comparisons test, levels of significance=P<0.05 (*), P<0.01 (**), P<0.001 (***), P<0.0001 (****). Additional information in **Supplementary file 1**.

The online version of this article includes the following source data for figure 5:

**Source data 1.** Datasets of thigmotaxis and swim speed analysis for **Figure 5C, D**.

Fluorescent stereomicroscopy recordings from living fish confirmed the results of PC quantification from fixed tissue. At 2 weeks post-treatment (wpt) control PC-ATTAC fish displayed a strong and continuous red fluorescence from the dorsal surface of their CCe, while in PC-ablated specimens, fluorescence was faint at best, or completely absent (**Figure 6—figure supplement 1A, B**). Strikingly, at 4 and 12 mpt a robust cerebellum-derived fluorescence had reappeared covering an increasing area of the CCe (**Figure 6—figure supplement 1C**). This fluorescence gradually converged to maximum intensity values obtained from adult controls (**Figure 6—figure supplement 1A, B**), suggesting a significant PC regeneration. The observation of EGFP fluorescent cells in the CCe of Tg(*ptf1a*:EGFP) adult fish showed that a source of progenitor cells for PCs is still present at advanced adult ages (**Figure 6—figure supplement 2A–C**). Furthermore, in the cerebellum of adult double transgenic Tg(*ptf1a*:EGFP) × Tg(PC-ATTAC) zebrafish, at least a few differentiated PCs (RFP positive) showed remaining GFP fluorescence derived from *ptf1a*:GFP progenitors providing evidence of *ptf1a*-expressing cells as progenitors of PCs also in the adult cerebellum (**Figure 6—figure supplement 2D, E**).

PC quantification on sagittal sections by immunohistochemistry against TagRFP-T and the PC-specific epitope ZebrinII (**Figure 6F–K**) confirmed that 2 weeks after PC ablation nearly any PC had survived in the adult CCe. Only few remaining PCs were preferentially located in caudal cerebellar regions (**Figure 6F, I**). At 4 mpt ZebrinII-expressing PCs with elaborate dendritic trees were distributed throughout a reappearing PC layer (**Figure 6D, G, J**) amounting to 28% of PCs found in controls (**Figure 7A, B**). Finally, at 12 mpt in vivo fluorescent stereomicroscopy and immunohistochemistry revealed that gaps between PCs had disappeared and ZebrinII-positive PCs were evenly distributed throughout a continuous PC layer (**Figure 6E, H, K**). Yet, the size of the PC population 1 year after ablation comprised to 30% in numbers compared to solvent controls (**Figure 7A, B**), which was not due to an overall growth reduction, as the average body length in both fish groups was nearly identical (**Figure 6—figure supplement 1D**). Of note, PC axon bundles projecting directly to the vestibular nuclei (**Knogler et al., 2019**; **Namikawa et al., 2019b**), which were completely degenerated at 2 wpt, were reestablished at 4 mpt and formed a prominent cerebello-vestibular tract at 12 mpt (**Figure 6—figure supplement 3A–F**).

These findings demonstrate that degeneration of the entire PC population in the adult zebrafish cerebellum does not represent an irreversible loss of these principal cerebellar neurons as previously thought. Instead, the adult zebrafish cerebellum maintains an impressive capability to regenerate significant numbers of PCs counteracting degeneration.

## Adult PC degeneration and regeneration follow a characteristic topographical pattern

With an average of 8% PC survivors, the caudal area of the adult CCe was more resistant to induced apoptosis than the rostral area, in which PC survival was sparse. When PC regeneration in sagittal sections of the CCe was evaluated based on morphological landmarks, medial and caudal regions of the PC layer were more efficient in replenishing lost PCs compared to rostral areas, irrespective of subdividing the CCe into two or three areas (**Figure 7—figure supplement 1A–C**). This patterned PC degeneration and regeneration may reflect the subdivision of the PC layer into different functional subcompartments along the rostro-caudal axis (**Knogler et al., 2019**; **Matsui et al., 2014**), and future elucidation of the underlying mechanisms promise to reveal important insights into PC maturation processes.

Recently, four different PC subtypes have been distinguished based on soma size correlating with a different dendrite morphology and physiology, but not being segregated with respect to their

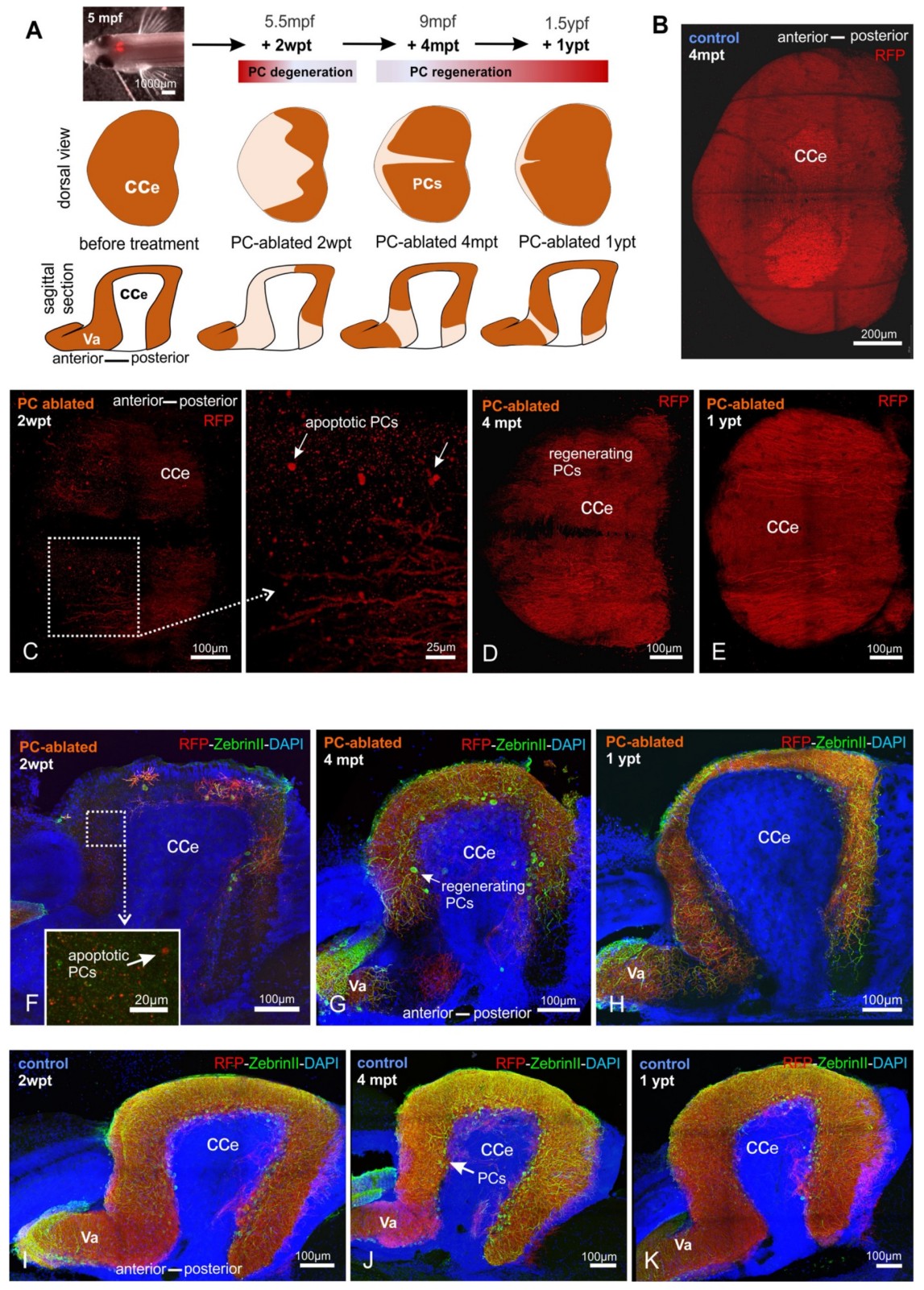

**Figure 6.** Monitoring of the Purkinje cell (PC) layer after induced PC ablation in adults. (**A**) In vivo stereomicroscopy showing tagRFP-T fluorescence in the PC layer, and illustration of the time course of fluorescence recovery after PC ablation in adults (at 5 mpf) monitored until 1 ypt. Representative confocal images of whole mount cerebelli from dorsal view (**B–E**) and sagittal vibratome sections after immunostaining with the antibodies anti-tagRFP and anti-ZebrinII (**F–K**), comparing the cerebellum in ablated fish (endoxifen treated; **C–E, F–H**) vs control group (dimethyl sulfoxide [DMSO] treated;

*Figure 6 continued on next page*

*Figure 6 continued*

**B, I–K**). Arrows in C, F point out apoptotic bodies. Equivalent results were observed in two additional independent ablations in adult cerebelli (data not shown).

The online version of this article includes the following source data and figure supplement(s) for figure 6:

**Figure supplement 1.** Imaging of the cerebellum in vivo, and quantification of the Purkinje cell (PC) area, body length, and swimming after PC ablation in adult zebrafish; related to *Figures 6 and 7*.

**Figure supplement 1—source data 1.** Measurements of maximum intensity of RFP fluorescence for *Figure 6—figure supplement 1B*, area size determinations of Purkinje cell (PC) layer for *Figure 6—figure supplement 1C*, adult zebrafish body size measurements for *Figure 6—figure supplement 1D*, mean swim speed and total distance values for behavioral analysis for *Figure 6—figure supplement 1E*.

**Figure supplement 2.** Distribution of *ptf1a*-expressing progenitors in the adult cerebellum (**A–C**).

**Figure supplement 3.** Degeneration and recovery of cerebello-vestibular tract after adult Purkinje cell (PC) ablation (A-F), related to *Figures 6 and 7*.

location in the PC layer (*Chang et al., 2020*; *Figure 7—figure supplement 1 E*). By measuring the size of PC somata all PC subtypes could be identified during de- and regeneration. Yet, compared to PCs in controls, apoptosis-resistant and regenerating PCs contained on average a larger soma size in all areas of the CCe, independent of subdividing the PC layer into two or three rostro-caudal compartments (*Figure 7—figure supplement 1D*). This could be due to a higher physiological demand on the few surviving PCs. Alternatively, PCs of the largest subtype I with a soma diameter larger than 10 µm are more resilient to apoptosis induction, and indeed in all rostro-caudal cerebellar compartments PCs with a large soma diameter (>10 µm) were significantly overrepresented at 2 wpt, which was maintained but less pronounced during regeneration up to 1 year (*Figure 7E*). Noteworthy, the anterior CCe that is most sensitive to apoptotic PC ablation, displayed the largest percentage of PCs with a soma diameter above 10 µm, ranging on average between 83% and 92% of the entire PC content (area 1, *Figure 7E*). These findings suggest that strong adapting subtype I PCs are more resilient to apoptotic cell death induction compared to PC subtypes II–IV. To address if such differences in PC subtype regeneration are already present during larval stages, we reanalyzed the cerebellum of zebrafish larvae after PC ablation. Indeed, based on soma size measurements, PCs with larger somata were overrepresented in the cerebellum with regenerating PCs compared to PCs in control larvae (*Figure 7—figure supplement 2*) suggesting that PC subtypes inherently possess different regeneration abilities already during PC differentiation stages.

## Reestablished cerebellar function by adult PC regeneration

Functional PC regeneration implies the restoration of behavior caused by adult PC loss known to result in a compromised exploration (*Buchberger et al., 2021*; *Elsaey et al., 2021*). In the novel tank test (NTT), adult zebrafish initially remain close to the bottom, but progressively start to explore the new environment (*Egan et al., 2009*; *Fontana et al., 2021*). Shortly after ablation despite the massive PC loss, zebrafish showed no significant difference neither in their mean swim speed nor in the total distance traveled compared to solvent-treated siblings (*Figure 6—figure supplement 1E*). This confirmed previous findings that zebrafish can largely relinquish PCs for locomotive control under non-strenuous conditions. In contrast, specimens with acute PC degeneration (2 wpt) as well as during PC regeneration (4 mpt) hardly ever entered the upper half of the new tank and displayed a significantly reduced exploratory behavior compared to controls (*Figure 7F, G*). Yet, in their home tanks, these fish fed successfully under continuous water flow. Importantly, after 1 year regeneration period this difference in exploratory behavior between PC-ablated and control fish had disappeared, demonstrating in this context the functional recovery of the ablated PC layer (*Figure 7G*), despite the lower total PC content compared to age-matched controls (*Figure 7A, B*). As PCs in the rostral area of the CCe are associated with regulation of non-locomotor behavior such as socio-emotional behavior, these findings suggest that also the less efficient PC regeneration in the adult rostral CCe is sufficient to regain proper control over exploratory behavior.

## Discussion

Acute invasive injuries have demonstrated the ability of the teleostean cerebellum including zebrafish to regenerate damaged neuronal structures by proliferation, migration and differentiation of neuronal

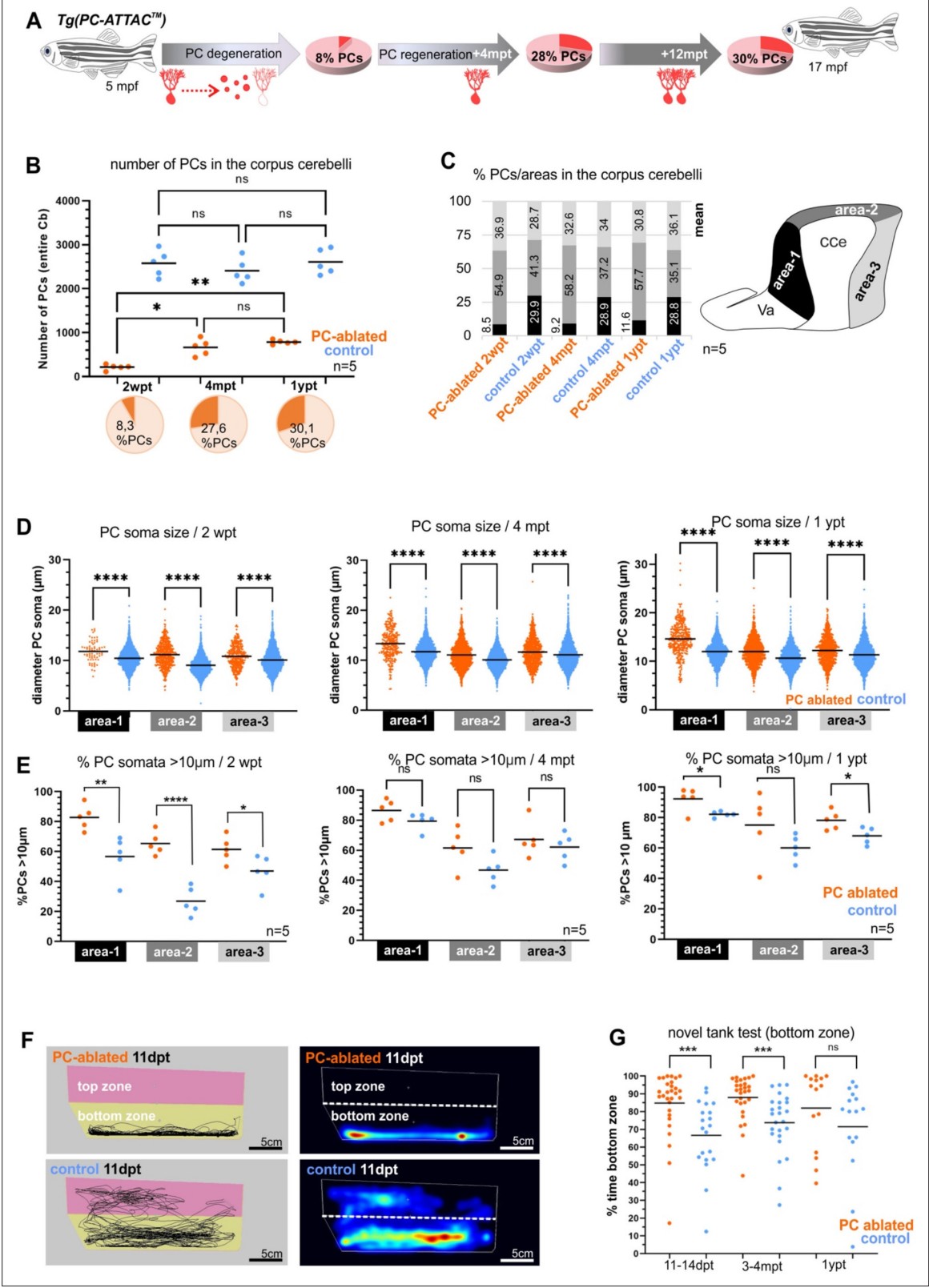

**Figure 7.** Quantitative analysis of Purkinje cells (PCs) and behavior test during degeneration and recovery after induced ablation in adults. (**A**) Illustration of the percentage of PC layer replenishment during degeneration and regeneration phases. (**B**) Number and percentage of PCs after induced PC ablation monitored until 1 ypt. (**C**) Subdivision of the corpus cerebelli (CCe) into rostral, dorsal and caudal areas (areas 1–3), and the respective proportion of PCs in each area. (**D**) Quantification of the diameter of PC somata in the different areas of the CCe. (**E**) Percentage of PCs with somata

*Figure 7 continued on next page*

*Figure 7 continued*

larger than 10 μm from the total amount of PCs per area per brain. (**F**) Representative images of swim tracks and heat map of location of adult zebrafish in the novel tank test during 6 min. (**G**) Percentage of time spent in the bottom zone. The data correspond to the results from three independent PC ablations that were pooled. The cellular quantification in all graphs represent PCs from the entire CCe of 5 fishes per group per time point, as average from the whole PC population of each brain (**B, C, E**), or at single-cell level that were pooled (**D**). Statistical information: statistical method=analysis of variance (ANOVA) test for multiple groups comparison (ordinary one-way ANOVA- followed by Sidàks multiple comparison test), or Student's t-test for two groups comparison (unpaired t-test two tailed for parametric, or Mann-–Whitney two tailed for no parametric data), levels of significance=P<0.05 (*), P<0.01 (**), P<0.001 (***), P<0.0001 (****). Additional information in *Supplementary file 1*.

The online version of this article includes the following source data and figure supplement(s) for figure 7:

**Source data 1.** Purkinje cell (PC) quantification for *Figure 7A-C*, PC somata measurements for *Figure 7D*, quantification of PC subtypes für *Figure 7E*, and exploration behavior in the novel tank test assays for *Figure 7G*.

**Figure supplement 1.** Quantification of patterned Purkinje cell (PC) death and recovery throughout the antero-posterior axis of the corpus cerebelli (CCe), related to *Figure 7*.

**Figure supplement 1—source data 1.** Distribution values of Purkinje cells (PCs) for *Figure 7—figure supplement 1B, C*. Measurements of PC diameters for *Figure 7—figure supplement 1D*.

**Figure supplement 2.** Diameter of Purkinje cell (PC) somata in larvae after PC ablation.

**Figure supplement 2—source data 1.** Quantification of average Purkinje cell (PC) somata diameter for *Figure 7—figure supplement 2B*, and quantification of PC somata of individual PCs in larvae for *Figure 7—figure supplement 2C*.

progenitors (*Kaslin et al., 2017*; *Zupanc and Zupanc, 2006*). With respect to PCs, regeneration in larvae has been deduced from the reappearance of PCs in injured cerebellar tissue. The proper physiological maturation of PCs and their contribution to cerebellum-controlled behavior has not been validated though, posing the question if reappearing PCs in larvae can be considered as a proper replacement for damaged PCs. Furthermore, acute injury studies consistently established that zebrafish PCs follow the human paradigm: while PCs in the developing and maturing cerebellum can still be replaced, the adult cerebellum has lost the ability to regenerate PCs. This change in regenerative potential has been explained by the exhaustion of a suitable PC progenitor cell type in the adult zebrafish cerebellum (*Kaslin et al., 2017*; *Kaslin et al., 2013*).

Acute injuries, which may represent only minor cases of cerebellar damage, have a massive impact on the surrounding neural tissue and are either locally restricted, as in the case of stab wounds, or lead to the destruction of the entire cerebellar architecture, as in the case for hemisphere removal or blunt force damage (*Fleisch et al., 2011*; *Hentig et al., 2021*; *Kaslin et al., 2017*). More commonly instead is the death of specific neuronal cell types and in particular PCs in the case of neurodegenerative diseases, for which PC regeneration has not been addressed in zebrafish so far. Therefore, the generalized view that zebrafish are unable to regenerate adult PCs needed to be reinvestigated by cell type-specific approaches and importantly non-invasive procedures.

We have readdressed this open question of PC regeneration with the help of the PC-ATTAC model, in which PCs can be ablated specifically by inducing apoptosis and reappearing PCs can be monitored by fluorescent protein expression (*Weber et al., 2016*). Using this model, we here confirmed that after depleting nearly the entire larval PC population, PCs regenerate to their full extent. The first regrowing PCs occurred within 5–7 days and were derived from their natural progenitors expressing the transcription factor *ptf1a*. Interestingly, this replacement of PCs did not occur by a rapid wave of significantly increased progenitor cell proliferation, but instead by a continuation of cerebellum growth accompanied by a slow but steady replenishment of PCs in number, until a PC population indistinguishable in size from non-PC-ablated control fish was reached.

An explanation for the slow but steady regeneration and stability of the circuitry can be derived from our physiological studies. Attached cell patch-clamp recordings from PCs in non-ablated control specimens confirmed that zebrafish PCs mature physiologically quite rapidly within 2 days, and the entire PC population displays stable electrophysiological patterns already at 8 dpf, 5 days after the first PCs are born (*Hsieh et al., 2014*; *Knogler et al., 2019*). In PC-ablated specimens, reappearing PCs with a mature physiological pattern could be observed as early as 4 dpt, with only marginal electrophysiological differences to control PCs at 8 dpt (*Figure 3—figure supplement 1E–G*). This fast reestablishment of PC physiological patterns allows for a slow rate of PC regeneration. Obviously, a proportion of the PC population (about half at 10 dpt *Figure 3—figure supplement 1D*) is able to

recover quickly and generates sufficient output to reestablish physiological properties closely resembling those of a healthy PC population.

We showed that during the acute phase of PC apoptosis this loss of inhibitory input resulted in ECs, the direct PC efferences, in a significantly increased average firing frequency of 47% and highest burst frequency of 82% at 3 dpt. With the reappearance of PCs at 4 dpt, a reduction in EC firing frequency back to normal values at 10 dpt was observed. This suggests that reappearing PCs quickly reintegrate into the PC circuitry. Thus, larval PCs do not only regenerate in cell number, but likely establish proper synaptic contacts. Alternatively, ECs adapt by plasticity to the lack of PC input along the same time-course.

Interestingly, the pattern of physiological maturation appeared to be the same between wild-type and regenerating PCs. An initially increased complex to simple spike ratio was reduced during PC maturation from 5 to 7 dpf, similar to firing patterns recorded from regenerating PCs from 5 to 7 dpt. This alteration in complex to simple spike ratio has been suggested to be derived from multiple climbing fiber innervation of PCs that is subsequently reduced by pruning until the strongest climbing fiber input remains (*Hsieh et al., 2014*). Once proper genetic targeting of inferior olivary neurons is available, it will be an interesting question to compare climbing fiber innervation in wild-type and regenerating PCs with adaptations in electrophysiological dynamics.

A fast recovery of mature and stable physiological patterns in regenerating PCs predicts a similarly quick recovery of PC function in governing behavior. In zebrafish larvae, impaired PC function has been shown to result in eye movement defects by means of saccade performance during the OKR (*Matsui et al., 2014*; *Namikawa et al., 2019b*), to lead to decreased exploratory behavior in anxiety-related tests and to cause delayed recovery from conditioned fear response (*Elsaey et al., 2021*; *Koyama et al., 2021*; *Matsuda et al., 2017*; *Namikawa et al., 2019b*). Indeed, we observed such behavioral deficits during acute PC apoptosis at 2 dpt, but also the functional recovery for visuomotor behavior (OKR), locomotor activity (hyperactivity), and socio-emotional behavior (thigmotaxis), between 9 and 11 days post-ablation, when the PC layer is only partially replenished (below 50% PC regeneration). These findings prove that PCs selectively ablated in larvae by apoptosis regenerate by repopulating the PC layer of the cerebellum in adequate numbers accompanied by the re-establishment of proper physiological and functional control. Thus, larval PC regeneration can be considered a true functional regeneration.

Studies on acute cerebellum injuries have established that PCs lose this ability to functionally regenerate during juvenile and young adult stages at about 3 months post-fertilization (*Hentig et al., 2021*; *Kaslin et al., 2017*). However, localized injuries could be counteracted by the functional plasticity of remaining PCs nearby, which may be faster compared to neuronal replenishment, thereby eliminating the need for a regenerative response of PC progenitor cells. Large scale injuries instead may remove or damage the proper cellular environment for regeneration.

We have therefore reinvestigated the potential of PC regeneration in the adult zebrafish cerebellum in the context of PC-specific cell ablation clearly beyond young adult stages starting at 5 months post-fertilization. This approach more closely reflects a degeneration of the PC population, as it occurs in the many known human ataxias. Surprisingly, we observed a strong and extensive regenerative response reestablishing about 28% of the entire PC population within 4 months and 30% within 1 year. Moreover, PCs also reestablished direct efferent projections to the vestibular nuclei in the ventral hindbrain. Compared to larvae, both the process of PC ablation and PC death as well as the regeneration required extended time periods. This is likely due to both a deceleration of neurogenesis and a reduced progenitor population (*Kani et al., 2010*; *Kaslin et al., 2017*) as well as a compact organization of a fully differentiated cerebellum with a larger size in adults compared to larvae.

Behavioral tests revealed that PC regeneration in adults was accompanied by recovery of PC-mediated compromised exploratory behavior. Therefore, PCs do not only reappear, but also reestablish characteristic functions of a wild-type PC population. Furthermore, these findings suggest that about one third of the size of the adult wild-type PC population is sufficient to reestablish cerebellum-controlled behavior maybe eliminating a need for further PC regeneration. Clearly, the adult zebrafish cerebellum is capable of significant functional PC regeneration under conditions of cell type-specific PC loss. That not the entire adult PC population is reestablished may be due to the apparent lower ratio of *ptf1a*-expressing progenitor cells to differentiated PCs in adults compared to the larval cerebellum,

a lack of need for further regeneration, prolonged PC differentiation processes in a compact and mature cerebellar cortex or incipient aging influencing or counteracting PC generation.

A recently established zebrafish model with compromised adult PC functions displayed defects in both exploratory and locomotor behavior (*Buchberger et al., 2021*). Interestingly, between PC-ablated PC-ATTAC and control specimens speed of swimming and distances traveled were not significantly different, but only exploratory behavior. This is consistent with findings in a recently generated zebrafish model of PC degeneration (*Elsaey et al., 2021*), and both PC degeneration models have in common that loss of PCs is non-homogenous with a more extensive PC loss in the anterior adult CCe and more resilient survivors in posterior regions. Intriguingly, such a patterned PC degeneration has been found in mouse models of several human cerebellar disorders in which anterior cerebellar regions are stronger affected by PC degeneration than posterior ones (*Sarna and Hawkes, 2011*; *Sarna and Hawkes, 2003*). Moreover, regeneration rates differed similarly as PC replenishment occurred faster in posterior than in anterior cerebellar regions. These observations support the view that suggested a functional pattern in the zebrafish cerebellum with posterior regions of the CCe predominantly controlling locomotor functions, while anterior cerebellar regions are more involved in mediating non-locomotor behavior such as socio-emotional behavior (*Chang et al., 2021*; *Chang et al., 2020*; *Harmon et al., 2017*; *Knogler et al., 2019*; *Knogler et al., 2017*; *Matsui et al., 2014*).

In addition to regional differences, patterned PC death in mammalian cerebellar diseases is also dependent on PC subtypes (*Sarna and Hawkes, 2003*). The existence of different PC subtypes in the adult zebrafish cerebellum has recently been established based on physiological criteria correlating to differences in PC soma size (*Chang et al., 2020*). PCs with soma diameters larger than 10 μm (PC type I) were significantly overrepresented in our studies after acute PC ablation and during adult PC regeneration. A simple explanation for this finding is that PCs in a depleted PC population use the available space for increasing in size. Alternatively, remnant PCs grow in size because of plasticity processes with PCs trying to recover as many functions as possible. Another explanation may be that the observed difference in PC soma distribution reflects a difference in resilience of PC subtypes to neurodegenerative processes or a different regenerative potential among PC subtypes. Addressing these questions by establishing the proper technical tools for investigating the regional as well as the subtype differences in PC regeneration represent future rewarding experimental goals to reveal further the mechanistic concepts and limitations of PC regeneration in the adult cerebellum.

## Materials and methods

### Key resources table

| Reagent type (species) or resource | Designation | Source or reference | Identifiers | Additional information |
|---|---|---|---|---|
| Antibody | anti-tagRFP (rabbit polyclonal) | Evrogen, Moscow, Russia | Catalog number: AB233 | IF (1:1000) |
| Antibody | anti-zebrinII (mouse monoclonal) | Donated by R. Hawkes, Univ. of Calgary, Canada | N/A | IF (1:500) |
| Antibody | anti-GFP (chick polyclonal) | Aves Labs, Davis, CA, USA | GFP-1010 | IF (1:1000) |
| Antibody | anti-BrdU (rat monoclonal) | Biozol GmbH, Eching, Germany | Abcam 6326 | IF (1:200) |
| Antibody | Alexa goat anti-rabbit IgG (H+L) 568 (goat polyclonal) | Invitrogen Inc, Carlsbad, CA, USA | A11011 | IF (1:500) |
| Antibody | Alexa goat anti-mouse IgG (H+L) 488 (goat polyclonal) | Invitrogen Inc, Carlsbad, CA, USA | A11001 | IF (1:1000) |
| Antibody | Donkey anti-chicken IgY (igG) (H+L) FITC (donkey polyclonal) | Jackson ImmunoResearch, West Grove, PA, USA | 703-545-155 | IF (1:1000) |
| Antibody | Alexa goat anti-rat IgG (H+L) 488 (goat polyclonal) | Invitrogen Inc, Carlsbad, CA, USA | A11006 | IF (1:1000) |
| Chemical compound/ drug | cis/trans-4-Hydroxy-tamoxifen (4-OHT) | Sigma-Aldrich, St. Louis, MO, USA | H6278 | |
| Chemical compound/ drug | Endoxifen | Sigma-Aldrich, St. Louis, MO, USA | E8284 | |

*Continued on next page*

*Continued*

| Reagent type (species) or resource | Designation | Source or reference | Identifiers | Additional information |
|---|---|---|---|---|
| Chemical compound/drug | BrdU | Sigma-Aldrich, St. Louis, MO, USA | B50021-G | |
| Commercial assay, kit | EdU (Click-EdU Alexa Fluor 647 Imaging kit) | Thermo Fisher Scientific, Waltham, MA, USA | C10340 | |
| Chemical compound/drug | d-Tubocurare hydrochloride-pentahydrate | Sigma-Aldrich, St. Louis, MO, USA | T2379 | |
| Chemical compound/drug | Tricaine, MS-222 | Sigma-Aldrich, St. Louis, MO, USA | E10521 | |
| Chemical compound/drug | DAPI | Thermo Fisher Scientific, Waltham, MA, USA | 62247 | |
| Chemical compound/drug | Normal Goat Serum, NGS | Vector Laboratories, Burlingame, CA, USA | S-1000 | |
| Chemical compound/drug | Albumin Fraction V | Carl Roth GmbH, Karlsruhe, Germany | 80076.2 | |
| Chemical compound/drug | DMSO | AppliChem GmbH, Darmstadt, Germany | 67-68-5 | |
| Strain, strain background (*Danio rerio*) | AB wild-type zebrafish | N/A | | Fish facility TU-Braunschweig |
| Strain, strain background (*Danio rerio*) | Brass pigmentation mutant zebrafish | N/A | | Fish facility TU-Braunschweig |
| Genetic reagent (*Danio rerio*) | Tg[ca8:-FynTagRFP-T2A-Casp8-ERT2 (PC-ATTAC)][bz11] | (**Weber et al., 2016**) 10.1242/dev.122721 | | Fish facility TU-Braunschweig |
| Genetic reagent (*Danio rerio*) | Tg(–7.5ca8:EGFP)[bz12] | (**Namikawa et al., 2019b**) 10.1177/1179069519880515 | | Fish facility TU-Braunschweig |
| Genetic reagent (*Danio rerio*) | Tg(olig2:EGFP)[vu12] Tg(olig2:dsRed)[vu19] | (**Shin et al., 2003**) doi: 10.1023/B:MICS.0000006847.09037.3a. (**Almeida et al., 2011**) doi:10.1242/dev.071001 | | Fish facility TU-Braunschweig |
| Genetic reagent (*Danio rerio*) | Tg(ptf1a:EGFP) | (**Godinho et al., 2005**) doi:10.1242/dev.02075 | | Fish facility TU Braunschweig |
| Genetic reagent (*Danio rerio*) | Tg(gfap:EGFP)[mi2001] | (**Bernardos and Raymond, 2006**) doi:https://doi.org/10.1016/j.modgep.2006.04.006 | | Fish facility TU Braunschweig |
| Software, algorithm | Patchmaster Software | HEKA Elektronik GmbH, Reutlingen, Germany | | |
| Software, algorithm | IGOR Pro 6.37 | Wavemetrics Inc, Portland, OR, USA | | |
| Software, algorithm | Leica LAS X | Leica Microsystems GmbH, Wetzlar, Germany | | |
| Software, algorithm | Prism9 Graph Pad | GraphPad Software, San Diego, CA, USA | | |
| Software, algorithm | Excel Microsoft | Microsoft Inc, Redmond, WA, USA | | |
| Software, algorithm | Free Imaging software FIJI | https://imagej.net | | |
| Software, algorithm | Ethovision Noldus software | Noldus Inc, Wageningen, The Netherlands | | |
| Software, algorithm | Corel Draw software | Corel Corporation, Ottawa, Ontario, Canada | | |

## Animal husbandry

All animals were raised and kept in the zebrafish facility in accordance with established practices adhering to the guidelines of the local government (***Aleström et al., 2020***; ***Westerfield, 2007***). AB wild-type or *brass* pigmentation mutant zebrafish were used for matings as indicated for each experiment. No selection criteria were used to allocate zebrafish of both sexes to any experimental group. Embryos and larvae were maintained in zebrafish 30% Danieau rearing medium (100% Danieau: 58 mM

NaCl, 0.7 mM KCl, 0.4 mM MgSO$_4$, 0.6 mM Ca (NO$_3$)$_2$, and 5 mM HEPES [4-(2-hydroxyethyl)-1-pipe razineethanesulfonic acid] pH 7.2) at 28°C. Larvae older than 7 days and adults were maintained in fish tanks according to standard protocols at 28°C, under light–dark conditions (simulating day–night cycle), and constant running water exchange. Stable transgenic lines used in this work include the Tg(ca8:-FynTagRFP-T2A-Casp8-ERT2)[bz11] line abbreviated (PC-ATTAC) (*Weber et al., 2016*) approved by LAVES of Lower Saxony, Germany (license AZ33.14-42502-04-068/07) and the reporter fish lines Tg(−7.5ca8:EGFP)[bz12], Tg(*olig2*:EGFP)[vu12], Tg(*ptf1a*:EGFP), and Tg(*gfap*:EGFP)[mi2001]. All experimental protocols for animal research were approved by governmental authorities of Lower Saxony, LAVES, (AZ33.19-42502-04-20/3593). All efforts were made to use only the minimum number of experimental animals necessary to obtain reliable scientific data.

## PC ablation

For PC ablation experiments in larvae (4–6 days post-fertilization), 4-OHT (cis/trans-4-hydroxy-tamoxifen, Sigma-Aldrich, St. Louis, MO, USA) at a concentration of 8 µM was added to the rearing medium for 18–20 hr. Since EtOH was used as solvent for 4-OHT, 0.4% EtOH was applied to the rearing medium of a control group for the same time period. On the next day, the medium was exchanged three times and the ablation rate of PCs was analyzed under a confocal laser scanning microscope (LSM). Ordinary ablation rates higher than 85–90% at day 2 post-treatment (dpt) compared to the control group were considered successful to use these larvae for further experiments.

To deplete PCs in adults, 5 months old zebrafish were incubated overnight in fish facility water supplemented with 4 µM Endoxifen (Sigma-Aldrich, St. Louis, MO, USA), an active metabolite of 4-OHT, which proved more efficient in adult PC ablations. Three consecutive treatments were performed each overnight at 28°C in the dark with resting and feeding periods during the day. Since dimethyl sulfoxide (DMSO) is used as solvent for Endoxifen, control groups were treated with 0.13% DMSO. Afterwards, the chemicals were washed out at least three times with fresh fish facility water and specimens were returned to the fish facility.

## BrdU and EdU incorporation assays

For proliferation studies in larvae, 10 mM BrdU diluted in Danieau's solution was added to the fish water after 4-OHT treatment in a 24-well plate (1 ml/well) for 24 hr, or up to 5 consecutive days in case of cumulative BrdU analysis. For BrdU/EdU double pulse chase experiment, incubation of BrdU was performed at 18 dpt for 24 hr at 10 mM, and incubation of EdU (*Click-IT Alexa Fluor 647 Imaging kit*, Thermo Fisher, C10340) at 22 dpt during 24 hr at 100 µM (stock solution 2.5 mg/ml diluted in DMSO) in a 6-well plate.

After the treatment, larvae were euthanized and the brains were isolated and processed for immunohistochemistry. The number of proliferating cells in each group (4-OHT/EtOH solvent control) within the PC layer of the cerebellum, and therefore excluding the caudal upper rhombic lip, was counted using the cell counter plugin of the image analysis software FIJI (see below). Student's *t*-test for two groups comparison (unpaired *t*-test two tailed) was applied.

## Brain tissue processing for immunohistochemistry and imaging

Brains of both, larvae and adults, were isolated and fixed by immersion in 4% paraformaldehyde (PFA) overnight at 4°C. For isolation of larval brains, fish were euthanized with an overdose of Tricaine and prefixed with PFA for 20–30 min, to allow for complete removal of the skin and brain isolation (using a minute pin 00 as dissection tool). After overnight fixation of the brain, PFA was washed out with an excess of phosphate-buffered saline (PBS), and the tissue was processed to a methanol/PBS series (30, 50, 70, and 100%) for gradual dehydration and final storage in 100% methanol at −20°C.

## Immunohistochemistry

For immunostaining, larval brains were rehydrated in a methanol–PBS series (70, 50, and 30%) and washed several times in PBS. Larval brains were processed for immunostaining as whole mount. After washing with PBS, antigen retrieval with 10 mM citrate buffer was achieved by heating samples in a water bath until boiling. After cooling to room temperature the brains were rinsed five times in PBS-T (PBS with 1% Triton X-100) for 5 min each, incubated in 100% precooled acetone for 15 min at −20°C to improve tissue permeabilization, followed by additional five PBS-T washing steps. Nonspecific

protein-binding sites were blocked in 5% normal goat serum in PBS-DT-I: PBS, 1% bovine serum albumin (BSA), 1% DMSO, 1% Triton X-100 for 1 hr at room temperature followed by primary antibody incubation in PBS-DT-II (PBS, 1% BSA, 1% DMSO, 0.3% Triton X-100) overnight at 4°C and additional 3 hr at room temperature. The primary antibodies: Rabbit anti-tagRFP (1:1000, Evrogen AB233, Moscow, Russia), mouse anti-ZebrinII (1:500, donated by Dr. Hawkes, University of Calgary, Canada), chicken anti-GFP (1:1000, Aves GFP-1010, Aves Labs, Davis, CA, USA), and rat anti-BrdU (1:200, Biozol GmbH 6326, Eching, Germany). Before anti-BrdU antibody application an additional incubation step with 2 N HCl was performed for 30 min at 37°C to denature the DNA for better anti-BrdU antibody accessibility. Subsequently brains were washed five times in PBS-T several times for 5 min each at room temperature with constant agitation and incubated with the secondary antibody diluted in PBS-DT-II overnight at 4°C. Secondary antibodies: Alexa goat anti-rabbit 568 (1:500, Invitrogen A11011, Carlsbad, CA, USA), Alexa goat anti-mouse 488 (1:1000, Invitrogen A11001, Carlsbad, CA, USA), donkey anti-chicken-FITC (1:1000, Jackson ImmunoResearch 703-545-155, West Grove, PA, USA), and goat anti-rat 488 (1:1000, Invitrogen A-11006, Carlsbad, CA, USA). For DAPI (4',6-Diamidino-2-phenylindole, 1 µg/ml) stained brains were incubated for 30 min at RT followed by several PBS-T washes. Finally, brains were thoroughly rinsed in PBS-T and PBS before imaging. In the case of EdU staining, EdU was detected after the incubation with primary antibodies (anti-tRFP, anti-BrdU) during the immunostaining procedure. The brains were washed twice with 3% BSA in PBS, washed with 0.5% Triton X-100 in PBS, washed twice with 3% BSA in PBS, added Click-IT reaction cocktail (1× Click iT EdU reaction buffer, CuSO$_4$, Alexa Fluor azide, 1× Click iT buffer additive; following the manufacturer's instructions), incubated for 30 min at RT in the dark, and finally washed with 3% BSA in PBS.

With respect to adult brains, immunostaining was carried out on 70 µm brain vibratome sections (Leica VT1000S, Leica Biosystems GmbH, Nusslsoch, Germany). Brains were pre-embedded in 1% low-melting agarose in PBS, and then embedded in 3% agarose. Vibratome settings: speed: 1 mm/s, vibration frequency: 75 Hz. The immunostaining was performed as described above, except for the antigen retrieval and acetone incubation. All incubations steps using adult brain sections were performed on floating sections, and eventually transferred to glass slides for confocal microscopy imaging.

## Microinjetion

Zebrafish embryos were injected at the one cell stage with plasmid DNA (25 ng/µl supplemented with 25 ng/µl mRNA encoding Tol2 transposase) (*Namikawa et al., 2019b*).

## Electrophysiological recordings

For single-cell patch-clamp recordings, zebrafish larvae were anaesthetized in a bath of **z**ebrafish **a**rtificial **c**erebro **s**pinal **f**luid (z-ACSF: 134 mM NaCl, 2.9 mM KCl, 2.1 mM CaCl$_2$, 1.2 mM MgCl$_2$, 10 mM HEPES, and 10 mM d-glucose at pH 7.8 [adjusted with 10 M NaOH]) containing 10 µM d-Tubocurare (Sigma-Aldrich, St. Louis, MO, USA). After complete loss of motor functions, the medium was removed and the larva was embedded in 2% low-melting agarose diluted in z-ACSF. Skin and skull tissue were then carefully removed with a sharp thin glass needle without damaging the underlying brain tissue to expose the cerebellum (see *Schramm et al., 2021* for a detailed method description). The immobilized larva within a square agarose block was immediately glued to a coverslip and transferred to the recording chamber, which was constantly perfused with oxygenated z-ASCF. Prior to electrophysiological recordings, larvae were adapted to the chamber without any disturbance for 10 min to allow acclimatization and/or stress reduction after the skull surgery. For attached cell recordings of PCs in 'voltage-clamp mode', electrodes with an inner resistance of 6–8 MΩ were pulled from borosilicate glass capillaries (Item#: 1B150F-4, ID: 0.84 mm, OD: 1.5 mm, WPI Inc, Sarasota, FL, USA) on a micropipette puller (Model PC-10, Narishige Inc, Tokyo, Japan) and filled with z-ACSF. PCs were identified by cytosolic EGFP expression Tg(−7.5ca8:EGFP)[bz12] (*Namikawa et al., 2019b*), and visualized with a fluorescence microscope (SCOPE-II, Scientifica Ltd, Uckfield, UK) equipped with a CCD camera (C10600, Hamamatsu, Photonics Europe GmbH, Herrsching, Germany). Patch-clamp electrodes were controlled by motorized micromanipulators (Scientifica Ltd, Uckfield, UK) for precise targeting of PCs. To reduce the risk of a pipette blockade, constant pressure was applied. Recorded signals were amplified with a TECH ITC-18 (HEKA Elektronik GmbH, Reutlingen, Germany). All PCs

were clamped with a holding potential of 0 mV throughout the whole time of recording and in a loose patch configuration with seal resistance reaching between 40 and 200 MΩ. For every cell, all spontaneous events were recorded at a rate of 20 kHz for a time span of 100 s and then selected with a Gaussian filter at 1 kHz. All experiments were performed at room temperature (25°C). For reliable results and to guarantee vitality/healthiness and normal cerebellar activity, larvae were only patched until a maximum of 1 hr after skull opening. The recorded cells were equally distributed between both hemispheres and regional localization. Also, 5 cells per fish were considered as maximum threshold for a fish and after the fifth successfully patched/attached neuron the next larva was put in focus. At the end of the experiment, zebrafish were euthanized by immersion in 0.2% Tricaine.

For regeneration studies of PCs, double transgenic fish carrying the PC-ATTAC and the –7.5*ca8*:EGFP transgenes were created by crossing carriers of the two transgenic lines Tg(*ca8*:FMA-TagRFP-2A-casp8-ERT2)[bz11] and Tg(–7.5*ca8*:GFP)[bz12]. Double-positive larvae were screened at 4 dpf and PCs were ablated by standard procedures. Double transgenic larvae incubated in 0.4% ethanol for 20 hr served as controls. The ablation rate was monitored by imaging larvae from 0 to 2 dpt each day under a confocal LSM and cell number analysis was performed. If the ablation rate was stable among all larvae and at 2 dpt over 90% compared to controls of the same age, the experiment was considered successful and patch-clamp recordings and imaging was performed from 3 to 21 dpt (10–28 dpf).

For patch-clamp recordings of ECs fish of the reporter line Tg(*olig2*:EGFP)[yu12] (**Shin et al., 2003**) were crossed with transgenic carriers of the Tg(PC-ATTAC)[bz11] line. In the double transgenic offspring, ablation and cell number analysis were performed as described above. ECs were visualized by cytosolic GFP expression, driven by *olig2* regulatory elements. Patch-clamp recordings were also performed in the attached cell configuration described above with electrodes from the same capillaries but with an inner resistance of 8–10 MΩ.

Electrophysiological data were recorded with the Patchmaster Software from HEKA (HEKA Elektronik GmbH, Reutlingen, Germany) and afterwards analyzed offline with Igor Pro 6.37 (Wavemetrics Inc, Portland, OR, USA) with a program code written by Marcus Semtner (Charité, Max Delbrück Center, Berlin, Germany). The counting of complex spikes was performed manually, whereas simple spike counting was automated. By this algorithm-based analysis all simple spikes were marked by the program, complete identification of all simple spikes was confirmed by subsequent visual inspection. Only traces which were stable for a minimum of 75 s and showed consistent spike patterns with comparable amplitudes were considered successful and taken for the final data analysis. The datasets from all recordings were collected in Excel 2019 (Microsoft Inc, Redmond, WA, USA) and then further analyzed and graphically represented with Prism9 (GraphPad Software, San Diego, CA, USA). Statistical relevance was analyzed by two-way analysis of variance (ANOVA) with Šídák's multiple comparisons test.

## Behavioral experiments
### OKR in larvae

The setup for this visuo-motor behavior test consisted of a movie projector playing a stimulating movie with vertical black and white stripes moving horizontally in constant speed projected over 180° across a circular screen as previously reported (**Matsui et al., 2014**). Single fish were mounted in front of the screen, in the center of a 5 cm Petri dish. The eye movement response of the fish to the stimulus was recorded by a camera mounted to a stereomicroscope objective above the head of the specimens. For the quantification of OKR response, the rotation angle of the right eye with respect to the body axis was measured. A rotation in the direction of the stimulus corresponds to the slow movement phase (marked as positive degrees). The rotation in the opposite direction or return of the eye to the original position, corresponds to the fast movement phase or saccade (marked as negative degrees, rotations of more than 19.5° were counted as saccade). The larvae analyzed were individually embedded using 2% low-melting agarose diluted in Danieau's solution. In order to analyze free movements of the eyes, a window in the agarose around the head was gently cut out, and the dish was filled with Danieau's solution (preventing distortion of the vision, as well as to ensure normal breathing and comforting conditions to the larvae). After 10 s of adaptation to the stimulus, the eye response was recorded for 45 s with a camera recording at 4 frames per second (independent trials without adaptation period showed consistent results). Afterwards, the larvae were released from the agarose, returned to the home tank, and finally sacrificed at the end of the experiment. The eye

response was recorded (Leica Microsystems LasX software, Wetzlar, Germany) using a stereomicroscope (MZC3000G, Leica Microsystems, Wetzlar, Germany) that was equipped with a digital camera (DFC3000G, Leica Microsystems, Wetzlar, Germany). For more consistency of the test results, the OKR recordings were always carried out between 10 am and 5 pm with alternating test rounds of each group. The statistical test applied was unpaired Student's *t*-test for parametric or normal distributed data.

### Free swimming in larvae

The analysis of free-swimming using larvae was performed using a custom-built zebrafish box (Noldus Inc, Wageningen, The Netherlands), illuminated with infrared and regular bright field light. Specimens were automatically tracked with the Ethovision XT 12 software (Noldus Inc, Wageningen, The Netherlands). The larvae placed in a 15-cm Petri dish were transferred to the behavior test room (at 28°C) for acclimatization at least 2 hr before the experiment. Subsequently, larvae were placed in a 12-well plate (one larva per well; with approximately 3 ml fish facility water/well) and let alone for an adaptation period of 5 min. The tracking was performed over a time period of 6 min. In order to quantify thigmotaxis, the arena was subdivided into middle and edge zones by the analytical software. Additional parameters measured were swim speed and total distance traveled. The larvae were returned to their respective fish tanks, and sacrificed at the end of the experiment. For more consistency of the results, the test was always carried out between 2:30 pm and 5:30 pm, with alternating of each group for swimming test analysis. The statistical tests applied were: ANOVA test for multiple group comparison (ordinary one-way ANOVA followed by the Šídák's multiple comparisons test for parametric or normal distribution data or Kruskal–Wallis test for no-parametric data).

### NTT in adults

The NTT was performed with a custom-built zebrafish box (Noldus Inc, Wageningen, The Netherlands) illuminated with infrared and regular bright field light, and automatic tracking with the Ethovision XT 12 software (Noldus Inc, Wageningen, The Netherlands). The fish were placed into behavior test room (at 28°C) approximately 1 hr prior to testing for acclimatization. For the NTT, the swimming of the fish was recorded for 6 min without a pre-adaptation period to the novel tank. The fish tank used for the test had a volume of 1.5 l (Aquaneering Inc, San Diego, CA, USA). After finishing the tracking, adult fish were returned to their respective home tanks, and sacrificed at the end of the experiment. For the tracking, the arena was subdivided into upper and bottom zones by the analytical software. Due to some variability of the NTT behavioral assay, individual control fish that showed clear exploration behavior of at least 5% occupancy in the upper half of the tank were chosen for further analysis. Mann–Whitney two-tailed test for no parametric data was applied.

## Imaging and data analysis

For imaging, larvae were anesthetized with 0.015–0.02% Tricaine (Sigma-Aldrich, St. Louis, MO, USA) dissolved in 1% low-melting agarose/Danieau solution. Images of the larval and adult zebrafish cerebellum were acquired using an SP8 laser scanning confocal microscope (Leica Microsystems, Wetzlar, Germany) with a ×40 objective. Images were processed with the imaging and analysis Leica LasX software.

Additional analysis of data was carried out with following software: FIJI (for automated quantification from image data and quantification of the OKR, freeware at https://imagej.net) and Ethovision (for swimming behavior quantification, Noldus Inc, Wageningen, The Netherlands). The quantification of PCs in juvenile and adult cerebelli was carried out after vibratome sectioning and immunostaining with anti-tagRFP and anti-ZebrinII antibodies. Semi-automated counting of cells was performed with the cell counter plugin of the FIJI software, in which every cell is initially identified manually followed by their automated detection with a z-stack of images to avoid, that cells are counted twice. The total amount of PCs per brain corresponds to the sum of all the sections. In order to minimize the error of double counting the same PCs from consecutive sections, cutting thick (70 µm) vibratome sections was considered the most appropriate sectioning method.

The design of the figures was carried out with CorelDraw software (Corel Corporation, Ottawa, Ontario, Canada).

## Material availability

Further information and requests for resources and reagents should be directed to and will be fulfilled by the Lead Contact.

## Acknowledgements

We gratefully acknowledge Kazuhiko Namikawa, Jakob von Trotha, and Florian Hetsch for helpful advice and experimental support. We thank Marcus Semtner (Max Delbrück Center, Berlin) for providing us with access to the Igor code for the analysis of electrophysiological data, and Janine Fichtner for support in their graphical presentation. We thank Alexandra Wolf-Asseburg, Iris Linde, and Timo Fritsch for excellent technical support, and all members of the Köster group for critical discussions. This work was supported by the European Union (Horizon 2020 research and innovation program under the Marie Sklodowska-Curie actions Individual Fellowships H2020-MSCA-IF-2015, grant agreement no. 703961, to SPM), the Volkswagenstiftung (project HOMEO-HIRN, ZN3673, to RWK and JCM), and the Bundesministerium für Bildung und Forschung (BMBF; Era-Net NEURON II CIPRESS to JCM).

## Additional information

### Funding

| Funder | Grant reference number | Author |
|---|---|---|
| Horizon 2020 Framework Programme | H2020-MSCA-IF-2015 703961 | Sol Pose-Méndez |
| Volkswagen Foundation | HOMEO-HIRN | Jochen C Meier Reinhard W Köster |
| Bundesministerium für Bildung und Forschung | Era-Net NEURON II CIPRESS | Jochen C Meier |
| Volkswagen Foundation | ZN3673 | Jochen C Meier Reinhard W Köster |

The funders had no role in study design, data collection, and interpretation, or the decision to submit the work for publication.

### Author contributions

Sol Pose-Méndez, Conceptualization, Data curation, Formal analysis, Supervision, Funding acquisition, Investigation, Methodology, Writing – original draft, Writing – review and editing; Paul Schramm, Barbara Winter, Conceptualization, Data curation, Formal analysis, Investigation, Methodology, Writing – original draft, Writing – review and editing; Jochen C Meier, Conceptualization, Data curation, Formal analysis, Supervision, Funding acquisition, Methodology, Writing – review and editing; Konstantinos Ampatzis, Supervision, Funding acquisition, Writing – review and editing; Reinhard W Köster, Conceptualization, Supervision, Funding acquisition, Writing – original draft, Writing – review and editing

### Author ORCIDs

Paul Schramm (iD) http://orcid.org/0000-0002-0894-2348
Konstantinos Ampatzis (iD) http://orcid.org/0000-0001-7998-6463
Reinhard W Köster (iD) http://orcid.org/0000-0001-6593-8196

### Ethics

All experimental protocols for animal research were approved by governmental authorities of Lower Saxony, LAVES (AZ33.19-42502-04-20/3593). All efforts were made to use only the minimum number of experimental animals necessary to obtain reliable scientific data.

### Decision letter and Author response

Decision letter https://doi.org/10.7554/eLife.79672.sa1
Author response https://doi.org/10.7554/eLife.79672.sa2

# Additional files

## Supplementary files

• Supplementary file 1. Statistical data related to *Figures 1–7* and respective figre supplements. Mean, standard deviation, statistic test applied, '*n*' number, p value, and level of significance are indicated. The reference to the different figures and groups of the comparisons performed in each figure are highlighted in bold. Abbreviations: 4-OHT 4-hydroxy-tamoxifen, BrdU Bromodeoxyuridine, DMSO dimethyl sulfoxide, dpf days post-fertilization, dpt days post-treatment, EdU Ethynyl-2'-deoxyuridine, Endox Endoxifen, EtOH ethanol, gfap glial fibrillary acidic protein, mpf months post-fertilization, mpt months post-treatment, NTT novel tank test, OKR optokinetic response, PC Purkinje cells, PCL Purkinje cell layer, ptf1a pancreas-associated transcription factor 1a, Tg transgenic, wpt weeks post-treatment, WT wild-type, ypf years post-fertilization, ypt years post-treatment.

• MDAR checklist

## Data availability

Detailed numbers for statistics shown in the figures are provided in *Supplementary file 1*. All data generated or analyzed during this study are included in the manuscript and supporting files. Further information and requests for resources and reagents should be directed to and will be fulfilled by the Lead Contact.

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
