## [Editor Report]

Using different zebrafish lines in combination with morphological and functional analysis, the authors provide compelling evidence that ventricular progenitors retains the life-long ability to regenerate PCs. At larval stages the newly regenerated PCs form fully functional circuits that lead to normal behavior. In adult, PC regeneration is less efficient but sufficient to support exploratory behavior. This fundamental study resolves the controversial issue of whether adult PC regeneration is possible and demonstrates that newly formed PCs at larval and adult stages can form functional circuits that supports normal behavior.

---

## [Decision Letter]

**Decision letter after peer review:**

Thank you for submitting your article "Lifelong regeneration of cerebellar Purkinje neurons after induced cell ablation in zebrafish" for consideration by *eLife*. Your article has been reviewed by 3 peer reviewers, one of whom is a member of our Board of Reviewing Editors, and the evaluation has been overseen by Marianne Bronner as the Senior Editor. The following individual involved in the review of your submission has agreed to reveal their identity: Luis Rodrigo Hernandez-Miranda (Reviewer #3).

The reviewers agree that this is quite exciting work; however, they all coincide in two flaws that need to be addressed to convincingly support the conclusions of the study.

The first is related to cell proliferation. This is not exhaustively analysed and therefore it is unclear whether it does contribute to Purkinje neuron regeneration at all the stages analysed.

The second issue relates to the identity of the adult precursor cells, which should be established.

The reviewers raise a few other criticisms that deserve attention if possible. The manuscript will also benefit from a more careful structure and labelling of the figures, which is not always accurate.

*Reviewer #2 (Recommendations for the authors):*

The manuscript would profit from a better emphasis on important conclusions. One specific example is some of the titles in the Results section which are rather descriptive and miss the opportunity to give the reader the immediate conclusion of the coming section (for example Progenitor cells of regenerating PCs). In general, this manuscript has a lot of exciting, important findings, which feel sometimes hidden in a lot of descriptive phrasing.

Comments and questions:

1. Is functional recovery due to new PCs being generated or could remaining cells and circuits rewire?

Would inhibition of progenitor cell proliferation (either globally or specifically targeting ptf1a-positive progenitors) assess whether functional recovery is inhibited in that scenario and therefore indeed due to regeneration of PCs?

Alternatively, would a second tamoxifen treatment, which should ablate newly regenerated PCs, lead to a lack of functional recovery?

2. Regeneration of PV neurons does not reach control numbers in the adult cerebellum, what is the interpretation of that result?

Are the Ptf1a numbers in the adult cerebellum lower in relation to the ones in young animals, such that the ratio of progenitors to neurons is less than in larvae. Are there potentially too few progenitors to lead to full regeneration? Can this be quantified?

3. A GFP transgenic reporter (ptf1a:gfp) was used to determine that these cells give rise to Purkinje cells and that gfap positive radial-glia-like cells are not the source.

Can it be excluded that gfap-positive cells generate ptf1a-positive progenitors which in turn would give rise to PCs? In the pictures in figure 2, it seems that gfap- and ptf1a reporter-positive cells are located in very similar regions, could there even be double-positive cells? It would be important to determine their respective localization and potential co-localization.

*Reviewer #3 (Recommendations for the authors):*

1) The authors describe no change in proliferative cells (BrdU^+^; Supplementary Figure 1A-D) during the regeneration of Purkinje cells. Which area was chosen for these quantifications?. When I look at Supplementary Figure 1A, I would quantify 34 BrdU^+^ cells in the PC-ablated condition vs 15 BrdU^+^ cells in the control condition. Please delineate the area that was chosen for the quantifications.

In lines 148-150, the authors conclude "This showed that PC regeneration was not initiated by a burst in cell proliferation, but occurred continuously by slowly adding surplus PCs over time until the full PC population size was reestablished 6 months later". This sentence is not clear to me. Would this mean that proliferating cells remain proliferating over longer times than in the control condition?

I would recommend the authors carry out a small experiment to assess the cycle length of precursor cells in the ablated larval condition, that is by combining pulses of BrdU and EdU. This could explain why no difference in proliferating (BrdU^+^) cells is seen in the ablated condition, as one of two possibilities can occur during the regeneration of Purkinje cells: (a) precursors divide faster to compensate for the loss of Purkinje cells or (b) precursors keep cycling over prolonged times, that is over more days than in controls.

2) The authors define ventricular zone Ptf1a+ cells as the precursors for the regenerated larval Purkinje cells. This, in my view, is not surprising as ventricular zone Ptf1a+ cells are precursors of Purkinje cells in most animals that have a cerebellum. Here, I have a few suggestions for the authors which can help the reader of this work to better understand this work:

a) Could you please indicate, in the introduction, the temporal window of Purkinje cell production in Zebrafish?

b) Could you please describe, in the introduction, whether ventricular zone Ptf1a+ cells are the natural precursors of Purkinje cells in Zebrafish?

c) Could you please mention whether inhibitory interneurons emerge from ventricular zone Ptf1a+ cells, as is the case in mammals? This can also be discussed in the Discussion section.

3) Whereas the authors show that Ptf1a+ cells are the precursors for the regenerated Purkinje cells in the larval stage, the suggestion that these cells are the source of regenerated Purkinje cells in the adult stage must be tested in this work. In my view, the elucidation of this can be one of the most exciting aspects of this work.

Therefore, I would recommend the authors carry out an experiment similar to that described in Figures 2A-D. That is using the Tg(gfap:EGFP; for radial glial cells) and Tg(ptf1a:EGFP; for the Ptf1a+ progenitors) in the adult ablated PC-ATTACTM Zebrafish background, and quantified the number of double positive RFP/GFP cells in the regenerating Tg(gfap:EGFP);PC-ATTACTM and Tg(ptf1a:EGFP);PC-ATTACTM animals. I would suggest focusing at least on the first 4 months post-regeneration. I know that this is a 4-month experiment, but I think this experiment will significantly add to this work.

4) Please restructure the manuscript. It is very confusing the description of Figures in the text versus the actual labeling of the Figures. It took me a lot of time to understand that the provided Supplementary Figure 1 contains the panels described for Figure 1 —figure supplement 1A-D; Figure 2—figure supplement 1E; Figure 2—figure supplement 1F; etc. Please separate the Figures and label them clearly.

---

## [Author Response]

The reviewers have discussed their reviews with one another, and the Reviewing Editor has drafted this to help you prepare a revised submission.The reviewers agree that this is quite exciting work; however, they all coincide in two flaws that need to be addressed to convincingly support the conclusions of the study.

We are grateful for the smooth and balanced handling of our manuscript. We appreciate very much the understanding of *ELife* that due to a parental leave of both lead authors until October 2022, working on the revision of the manuscript was delayed. We have carefully read all concerns and have addressed them point by point also delineating the changes that were made to the carefully revised manuscript.

The first is related to cell proliferation. This is not exhaustively analysed and therefore it is unclear whether it does contribute to Purkinje neuron regeneration at all the stages analysed.

We have followed this advice and have added the suggested analysis of BrdU positive cells at a later stage of PC regeneration, which we answer in our response to the first question raised by reviewer 1. In addition, we have performed the dual BrdU/EdU-labeling experiment to address further upregulation of proliferation and a potential prolongation of cell cycle exit as suggested in the first comment of reviewer 3. We would like to refer the editor to our answers to these points and we hope that we could have clarified this concern.

The second issue relates to the identity of the adult precursor cells, which should be established.

We have also attempted to solve this question, and we now show that indeed in the adult cerebellum *ptf1a*:GFP expressing cells are progenitors of PCs. For this we have established double transgenic *ptf1a*:GFP/PC-ATTAC adult zebrafish in which we were able to identify red fluorescent PCs expressing with remnants of GFP expression. We explain these findings in our answer to question 3 of reviewer 3 and reveal these new data in a revised new Supplementary Figure 13.

To however trace *ptf1a*:GFP cells as progenitors of regenerating PCs in adults requires the establishment and validation of additional sophisticated transgenic strains, which in our opinion represents a whole scientific project on its own that extends beyond the current focus of the manuscript.

The reviewers raise a few other criticisms that deserve attention if possible. The manuscript will also benefit from a more careful structure and labelling of the figures, which is not always accurate.

We have read through all these points of concern and have carefully addressed each criticism. Please find our detailed answers below which we hope resolve of points of dispute.

Reviewer #2 (Recommendations for the authors):The manuscript would profit from a better emphasis on important conclusions. One specific example is some of the titles in the Results section which are rather descriptive and miss the opportunity to give the reader the immediate conclusion of the coming section (for example Progenitor cells of regenerating PCs). In general, this manuscript has a lot of exciting, important findings, which feel sometimes hidden in a lot of descriptive phrasing.

We thank the reviewer for these supportive comments. We have followed the advice by reformulating the headlines of the individual paragraphs of the result section into conclusions and we have significantly revised the entire manuscript accordingly.

Comments and questions:1. Is functional recovery due to new PCs being generated or could remaining cells and circuits rewire?

Based on the PC ablation and functional recovery in adults, we estimate that about 1/3 of the PC population is required to establish sufficient cerebellar circuitry for functional recovery in larvae. With ablation rates close to or above 90% of the PC population, the few remaining cells will not be able to reestablish a functional cerebellar cortex in larvae. In particular, since the cerebellar PC population is known to be subdivided in different functional compartments such variation in PC connectivity could not be covered by the few remaining PCs. Therefore, it is highly unlikely that rewiring of ablation-resistant PCs is sufficient to recover cerebellar functions, while the many regenerating PCs do not reestablish proper circuitry. Moreover, our electrophysiological data showing that all replenished PCs establish expected patterns of neurotransmission argue against such a scenario.

Would inhibition of progenitor cell proliferation (either globally or specifically targeting ptf1a-positive progenitors) assess whether functional recovery is inhibited in that scenario and therefore indeed due to regeneration of PCs?Alternatively, would a second tamoxifen treatment, which should ablate newly regenerated PCs, lead to a lack of functional recovery?

This is an interesting experimental suggestion, but it is technically not feasible. Global inhibition of cell proliferation will also affect PC afferent and efferent structures of first and higher orders making it impossible to discriminate between an impaired functional recovery caused by insufficient numbers of PCs or defects in differentiation along the circuitry.

For a PC progenitor specific inhibition of proliferation using genetics the best regulatory element available would be the *ptf1a*-enhancer. But this regulatory element drives expression as well in progenitors of eurydendroid cells, the first order efferents of PCs, and inhibitory interneurons as first order afferents of PCs not allowing one to discriminate between manipulating PC, EC and inhibitory interneuron progenitor proliferation.

Alternatively, would a second tamoxifen treatment, which should ablate newly regenerated PCs, lead to a lack of functional recovery?

Authors: We have followed this advice and performed this experiment. 10 days after the first Tamoxifen-induced PC ablation, when roughly 50% of the PC population was reestablished (49.1 %), a second Tamoxifen-treatment was performed reducing again the PC population to 6.6%. Within the following 10 days the PC population recovered to 64% compared to numbers of PCs in control larvae. This shows that PCs can repeatedly regenerate and also replace the newly regenerated PCs for another time.

To address the functional recovery of these PCs we performed electrophysiological recordings. Similar to the first ablation, the average frequency and highest frequency of PCs decreased significantly, while the ratio between complex to simple spikes increased during the acute degeneration phase (3 days post second treatment). With the second 10 days regeneration period the average frequency and complex to simple spike ratio recovered to values indistinguishable to PCs in non-ablated control larvae. This indicates that also after the second ablation repeatedly regenerating PCs recover to normal firing patterns.

Moreover, if an ablation resistant PC population existed and regenerating PCs remained non-functional, this ablation resistant fraction of PCs would have reestablished normal PC firing patterns by plasticity mechanisms during the first regeneration period. Consequently, the second ablation should not affect the physiological signature of these potential ablation resistant PCs anymore. Yet, our physiological measurements revealed clear physiological changes and patterns of recovery reminiscent to the first PC ablation and regeneration recovery. This is strong evidence against the existence of a small but ablation resistant PC population that reestablishes the function of the entire PC population merely by plasticity. We are therefore convinced that the repeated functional recovery of the large majority of the ablated PC population is driven by the incorporation and proper rewiring of newly added regenerating PCs.

We thank the reviewer for this constructive suggestion which we believe made our data about PC regeneration stronger and further convincing. We have therefore added these findings to the manuscript and display the corresponding data in a new Supplementary Figure (Figure 3—figure supplement 2).

2. Regeneration of PV neurons does not reach control numbers in the adult cerebellum, what is the interpretation of that result?Are the Ptf1a numbers in the adult cerebellum lower in relation to the ones in young animals, such that the ratio of progenitors to neurons is less than in larvae. Are there potentially too few progenitors to lead to full regeneration? Can this be quantified?

The numbers of *ptf1a*:GFP progenitors in adults are apparently lower than in larvae and juveniles as these undergo cerebellar development with a peak of PC generation. We have added this possible explanation to the discussion. In addition, in adults PC generation and differentiation occurs at a much slower pace. But even in adults one year of age *ptf1a*:GFP positive progenitor cells can be identified and seem to add PCs to the PC population. Hence there is no indication that the PC progenitor population is exhausted.

Functional recovery in adults is achieved when about 30% of the PC population has been re-established. At this level of recovery, the pressure for further regeneration is lifted, which may contribute to slowing down PC regeneration. In addition, PC differentiation in adults requires much longer time periods likely due to reduced differentiation signals, longer migration pathways and a more compact tissue that progenitor cells have to cope with. Furthermore, ongoing PC regeneration may be further influenced or counteracted by beginning aging-associated processes at 1.5-2 years in the nervous system. To provide explanations for the incomplete regeneration of the adult PC population, we have added these arguments to the discussion which reads now:

“Furthermore, these findings suggest that about one third of the size of the adult wildtype PC population is sufficient to reestablish cerebellum-controlled behavior maybe eliminating a need for further PC regeneration. Clearly, the adult zebrafish cerebellum is capable of significant functional PC regeneration under conditions of cell type specific PC loss. That not the entire adult PC population is reestablished may be due to the apparent lower ratio of *ptf1a*-expressing progenitor cells to differentiated PCs in adults compared to the larval cerebellum, a lack of need for further regeneration, prolonged PC differentiation processes in a compact and mature cerebellar cortex or incipient aging influencing or counteracting PC generation.”

3. A GFP transgenic reporter (ptf1a:gfp) was used to determine that these cells give rise to Purkinje cells and that gfap positive radial-glia-like cells are not the source.Can it be excluded that gfap-positive cells generate ptf1a-positive progenitors which in turn would give rise to PCs? In the pictures in figure 2, it seems that gfap- and ptf1a reporter-positive cells are located in very similar regions, could there even be double-positive cells? It would be important to determine their respective localization and potential co-localization.

The reviewer is right that we cannot exclude that *gfap*-positive cells are progenitor cells of *ptf1a*-expressing cells and we have stated that in our manuscript (line 182-183). We did not find GFP-expression of transgenic *gfap*:GFP carriers to colocalize with red fluorescent PCs of the PC-ATTAC strain. While this argues against *gfap*-positive cells to give rise to PCs, we cannot exclude that GFP expression is not sufficiently stable to still provide sufficient green fluorescence in differentiating PCs.

To address whether *gfap*-positive radial glia cells are progenitor cells of *ptf1a*:GFP positive PC progenitors, we undertook additional experiments which are displayed in a new Supplementary Figure 4 and we refer to them in the manuscript. Because *gfap*:GFP and *ptf1a*:GFP transgenic carriers express the same fluorescent protein (GFP), we could not make use of double transgenic larvae. We therefore used an anti-Brain lipid-binding protein antibody (Blbp) for immunohistochemistry, which detects almost all *gfap*-positive radial glia cells in the developing zebrafish cerebellum (90%). Anti-Blbp immunohistochemistry in *ptf1a*:GFP zebrafish larvae revealed only a small fraction of about 10% of *ptf1a*:GFP positive cells to coexpress Blbp and an even smaller fraction of *olig2*:GFP positive cells to coexpress Blbp, while none of the red fluorescent PC progenitors were found to coexpress Blbp. These findings argue that a small fraction of cells in the cerebellum expressing radial glia proteins in *ptf1a*:GFP cells likely contribute to the small fraction of *olig2*:GFP expressing oligodendrocytes of the cerebellum rather than to specific neuronal populations such as PCs.

While these data further argue for *ptf1a*:GFP but not *gfap*:GFP positive cells to act as progenitors of PCs, these data are still not unambiguously conclusive as a potential short half-life of Blbp would interfere with the identification of radial glia protein expression in the majority of *ptf1a*:GFP cells. The conclusive identification of radial glia gene expression in *ptf1a*:GFP and PC-ATTAC cells will require permanent fate mapping approaches such as Cre-recombinase mediated permanent reporter gene expression in *gfap*:GFP derivatives and require genetic tools which are currently not available.

Reviewer #3 (Recommendations for the authors):1) The authors describe no change in proliferative cells (BrdU^+^; Supplementary Figure 1A-D) during the regeneration of Purkinje cells. Which area was chosen for these quantifications?. When I look at Supplementary Figure 1A, I would quantify 34 BrdU^+^ cells in the PC-ablated condition vs 15 BrdU^+^ cells in the control condition. Please delineate the area that was chosen for the quantifications.

We agree that it is of help to demarcate the area chosen for quantification of cell proliferation. We have now added a dashed white line to Figure 1—figure supplement 2A outlining the respective areas used for counting BrdU^+^ cells, which is explained in the legend of this figure.

In lines 148-150, the authors conclude "This showed that PC regeneration was not initiated by a burst in cell proliferation, but occurred continuously by slowly adding surplus PCs over time until the full PC population size was reestablished 6 months later". This sentence is not clear to me. Would this mean that proliferating cells remain proliferating over longer times than in the control condition?I would recommend the authors carry out a small experiment to assess the cycle length of precursor cells in the ablated larval condition, that is by combining pulses of BrdU and EdU. This could explain why no difference in proliferating (BrdU^+^) cells is seen in the ablated condition, as one of two possibilities can occur during the regeneration of Purkinje cells: (a) precursors divide faster to compensate for the loss of Purkinje cells or (b) precursors keep cycling over prolonged times, that is over more days than in controls.

We have followed this helpful advice of this reviewer and have implemented into this study the questions of reviewer 1 to address cell proliferation in the PC regenerating cerebellum at around 3 weeks after PC ablation. A first pulse of BrdU-labeling was applied 18 days after the ablation followed by a pulse of EdU-labeling four days later. Subsequently, four days after the EdU labeling BrdU^+^-, and double labeled BrdU^+^/EdU+-cells also expressing tagRFP-T to confirm their recent differentiation into Purkinje cells in the cerebellum were quantified. The number of BrdU^+^ cells did not show a significant difference between PC-ablated and control larvae, the number of BrdU^+^/EdU+- Purkinje cells was much lower suggesting that most Purkinje cell progenitors underwent differentiation instead of dividing repeatedly. Yet, in PC-ablated specimens the numbers of BrdU^+^/EdU+- double positive Purkinje cells were significantly higher compared to controls. This suggests that in PC-ablated zebrafish PC progenitors add on average more Purkinje cells to the Purkinje cell layer by additional rounds of proliferation. The absolute numbers of such PC progenitors with additional rounds of proliferation were low, supporting our hypothesis that PC ablation does not elicit a strong proliferation response in Purkinje cell progenitors. Instead, additional Purkinje cells are added slowly over longer periods of time by a few progenitors differentiating later after additional rounds of proliferation. This explains why completing the full number of Purkinje cells equivalent to numbers in controls requires several months. These results are presented in a new Supplementary Figure (Figure 1—figure supplement 3) and have been added to the last paragraph of describing the results presented in Figure 1.

2) The authors define ventricular zone Ptf1a+ cells as the precursors for the regenerated larval Purkinje cells. This, in my view, is not surprising as ventricular zone Ptf1a+ cells are precursors of Purkinje cells in most animals that have a cerebellum. Here, I have a few suggestions for the authors which can help the reader of this work to better understand this work:a) Could you please indicate, in the introduction, the temporal window of Purkinje cell production in Zebrafish?

First Purkinje cells expressing PC specific genes such as carbonic anhydrase 8 or ZebrinII appear in the zebrafish cerebellum about 3 days postfertilization (dpf). And this first wave of PC generation lasts until 6dpf when a sufficiently rudimentary PC population has been established and a first plateau in PC generation has been reached (Bae et al., 2009; Hamling et al., 2015; Namikawa et al., 2019). Subsequently, fewer PCs are added slowly over time to achieve a slow continuous increase in PC numbers aligned with larval growth. We have now added this information to the second paragraph of the introduction.

b) Could you please describe, in the introduction, whether ventricular zone Ptf1a+ cells are the natural precursors of Purkinje cells in Zebrafish?c) Could you please mention whether inhibitory interneurons emerge from ventricular zone Ptf1a+ cells, as is the case in mammals? This can also be discussed in the Discussion section.

We agree that such information is helpful to the reader who is not familiar with zebrafish cerebellar development. We have therefore added the following information to the introduction:

“Furthermore, clarification of the developmental origin and differentiation program of zebrafish cerebellar neurons, their physiology and functional contribution to locomotor control, motor learning and socio-emotional behavior has been revealed. For example, like in mouse, Purkinje cells, together with inhibitory interneurons and eurydendroid cells – deep nuclei neuron equivalents – in zebrafish have been shown to arise from progenitor cells derived from the cerebellar ventricular zone expressing the *pancreas associated transcription factor 1a* (*ptf1a*) (Kani et al., 2010; Kaslin et al., 2013). In zebrafish, a first wave of postmitotic PCs appears from 3 to 6 days postfertilization (dpf), afterwards fewer PCs are added slowly over time to achieve a slow continuous increase in PC number aligned with larval growth (Bae et al., 2009; Hamling et al., 2015; Kazuhiko Namikawa et al., 2019). Controversial reports exist about whether in the adult zebrafish cerebellum new Purkinje cells are still generated (Kani et al., 2010; Kaslin et al., 2013, 2009). Together, these studies have laid a solid foundation for regeneration surveys of cerebellar neurons and PCs in particular (Chang et al., 2021b, 2020; Harmon et al., 2017; Kidwell et al., 2018; Knogler et al., 2019, 2017; Koyama et al., 2021; Markov et al., 2021; Matsuda et al., 2017; Matsui et al., 2014; Rieger et al., 2009; Volkmann et al., 2008).“

Given the current lack of genetic tools to unambiguously trace *ptf1a*-expressing progenitor cells in the late larval and adult zebrafish cerebellum as well as the limited space in the discussion, we prefer not to elude too much about the differentiation potential of these progenitors into cerebellar interneurons as we have not investigated this in detail in the present study.

3) Whereas the authors show that Ptf1a+ cells are the precursors for the regenerated Purkinje cells in the larval stage, the suggestion that these cells are the source of regenerated Purkinje cells in the adult stage must be tested in this work. In my view, the elucidation of this can be one of the most exciting aspects of this work.Therefore, I would recommend the authors carry out an experiment similar to that described in Figures 2A-D. That is using the Tg(gfap:EGFP; for radial glial cells) and Tg(ptf1a:EGFP; for the Ptf1a+ progenitors) in the adult ablated PC-ATTACTM Zebrafish background, and quantified the number of double positive RFP/GFP cells in the regenerating Tg(gfap:EGFP);PC-ATTACTM and Tg(ptf1a:EGFP);PC-ATTACTM animals. I would suggest focusing at least on the first 4 months post-regeneration. I know that this is a 4-month experiment, but I think this experiment will significantly add to this work.

We agree with the reviewer that this is an interesting question to address. We have therefore investigated adult zebrafish *ptf1a*:GFP/PC-ATTAC double transgenics and we were able to reveal that also in adults of about 6 months of age double fluorescent cells exist in the PC layer therefore showing for the first time that *ptf1a*:GFP progenitors cells continue to give rise to PCs also during adult stages. These findings have been added now to the new Supplementary Figure (Figure 6—figure supplement 2).

To address this same question for regenerating PCs is technically far more challenging than assumed by the reviewer and has to await the advent of additional genetic tools. As shown in our studies the differentiation of regenerating progenitor cells into Purkinje cells occurs much more slowly compared to larval stages. Progenitors of Purkinje cells downregulate their expression of *ptf1a* during the course of differentiation. Therefore, GFP expression will be lost in these progenitors over time, and it remains questionable whether GFP-stability will allow for observing double fluorescent regenerated Purkinje cells. Only in the case of a positive outcome, this experiment will be interpretable. In case, none GFP-positive regenerated PC-ATTAC Purkinje cells in the *ptf1a*-GFP background could be observed, it cannot be excluded that *ptf1a+*-cells do not represent the progenitor cells of regenerating Purkinje cells, as these progenitors could have lost GFP fluorescence prior to activating tagRFP-T expression. In order to address whether *ptf1a*+-expressing cells are the progenitor cells of regenerating Purkinje cells in the adult cerebellum a permanent labeling method for *ptf1a*+ cells e. g. by a Cre-recombinase-mediated approach would be needed. Such a study requires generation and validation of stable transgenic zebrafish strains, and we hope the reviewer agrees with our opinion that these fate mapping studies represent a full project of their own, which is beyond the scope of the current manuscript.

4) Please restructure the manuscript. It is very confusing the description of Figures in the text versus the actual labeling of the Figures. It took me a lot of time to understand that the provided Supplementary Figure 1 contains the panels described for Figure 1 —figure supplement 1A-D; Figure 2—figure supplement 1E; Figure 2—figure supplement 1F; etc. Please separate the Figures and label them clearly.

We thank the reviewer for this advice, and we agree that the naming of supplementary material is cumbersome. But this labeling of supplementary figures adheres to the format required by *eLife* and our initially submitted manuscript was returned to us demanding that we stick to these figure labeling rules. Yet, in the revised manuscript we have more clearly separated the supplementary material and hope that this will facilitate to relate supplementary material to main figures.